# Sequestration and suppressed synthesis of oncogenic HMGA1 using engineered adenoviruses decreases human pancreatic and breast cancer cell characteristics

Md. Sharif Hasan[1], Shuisong Ni[1], Fatema B. Kamal[2,3], Megan F. Blossey[1], Eian Vargas[1], Margaret B. Bogomolny[1], Trang Dinh[1], Michael A. Kennedy[1]*

**1** Department of Chemistry and Biochemistry, Miami University, Oxford, Ohio, United States of America, **2** Mailman School of Public Health, Columbia University, New York, New York, United States of America, **3** National Center for HIV, Viral Hepatitis, STD and TB Prevention, ORISE/CDC, Atlanta, Georgia, United States of America

* kennedm4@miamioh.edu

## Abstract

HMGA1, an architectural transcription factor that plays a crucial role in tumorigenesis, chemotherapy resistance and cancer stem cell transformation in many human cancers, is intrinsically disordered and cannot be targeted by conventional small molecule drug therapy. While HMGA1 is required and essential for normal growth and development, HMGA1 expression occurs at very low levels in normal healthy adult cells. In contrast, HMGA1 is expressed at very high levels in many different types of human cancer cells. Since HMGA1 cannot be targeted using conventional small molecule drug therapy, alternative approaches are needed to target HMGA1 in new cancer therapies. Here, we explored the use of serotype 5 adenoviruses (Ad5) engineered 1) to sequester overexpressed HMGA1 in cancer cells using an HMGA1 hyper binding site (HBS) inserted into the Ad5 genome and 2) to suppress HMGA1 synthesis in cancer cells by incorporating exogenous genes into the Ad5 genome that encode an artificial HMGA1 cis-antisense transcript (AAT) and that encode a gene to express an HMGA1-targeted shRNA transcript (shRNA). The three engineered Ad5s were tested in MiaPaCa-2, PANC-1 and BxPC-3 human pancreatic cancer cell lines and in the ZR-75 human breast cancer cell line. Cancer cell viabilities and cell migration capability decreased by ~50–75% with HBS viruses and by 25–50% for shRNA and AAT viruses. Anchorage-independent migration capacity decreased by 60–70% with all three HBS, shRNA and AAT viruses. HMGA1 mRNA transcripts levels varied from 100 to 300 copies per cell in untreated cells and these levels were not significantly affected by treatment with the HBS and shRNA viruses, however the HMGA1 mRNA levels increased by ~3-fold upon AAT virus treatment. HMGA1 protein levels decreased in the range of 40, 50 and 70% with shRNA, AAT and HBS viruses, respectively. The HBS virus designed to sequester HMGA1 proved most effective

**Data availability statement:** All relevant data are within the manuscript and its Supporting Information files.

**Funding:** The author(s) received no specific funding for this work.

**Competing interests:** The authors have declared that no competing interests exist.

overall in suppressing HMGA1 oncogenic activity in these *in vitro* cell-based studies compared to the AAT and shRNA viruses.

## Introduction

HMGA1, a member of the High Mobility Group A (HMGA) family, is an architectural transcription factor that binds genomic DNA in the narrow minor groove structure of adenine-thymine (AT)-tracts [1]. In doing so, HMGA1 can remodel chromatin structure, allowing other transcription factors to bind to DNA and regulate gene expression [2]. Although highly expressed during embryogenesis, HMGA1 expression is nearly null in healthy adult cells [3]. In sharp contrast, HMGA1 is overexpressed at very high levels in many kinds of human cancer cells [2–4]. The links between HMGA1 over-expression with tumor growth and development, drug resistance, poor prognosis and direct role in cancer development are well established [3–5].

Elevated HMGA1 levels play a critical role in pancreatic cancer tumorigenesis [6]. Pancreatic cancer remains a devastating disease with poor survival statistics and there is a dire need for improved diagnostic techniques for early detection and for development of novel therapies [7, 8]. Previously, we demonstrated that transfection of human pancreatic cancer cells with DNA aptamers containing HMGA1 binding sites could significantly reduce human cancer cell viability [9]. Building on this observation, we developed an adenoviral vector containing exogenous DNA with an HMGA1 hyper binding site (HBS) capable of delivering multiple HMGA1 sequestration sites directly to cancer cells. We demonstrated that this approach significantly decreased viability of multiple pancreatic and liver cancer cells [10]. Using a mouse model, we demonstrated that the engineered adenovirus vectors caused no measurable biotoxicity and a biodistribution analysis indicated that they did not cross the blood brain barrier following injection into the pancreas [11]. Motivated by our prior results illustrating that HMGA1 sequestration reduced cancer cell characteristics, we engineered 1) an adenovirus vector containing an exogenous gene encoding an HMGA1 artificial antisense transcript (AAT) and 2) an adenovirus vector containing an exogenous gene encoding an HMGA1 shRNA transcript (shRNA), both designed to suppress HMGA1 translation and to block HMGA1 protein synthesis. Generation of these two new engineered adenovirus vectors was based on the hypothesis that suppression of HMGA1 synthesis in cancer cells would potentially provide stronger anti-cancer activity compared to infection with adenoviral vectors exhibiting HMGA1 sequestration capability alone. Recent advances in viral vector engineering have demonstrated the power of such tools for studying gene therapy approaches and its application in oncology research [12].

Here, the efficacies for reducing cancer cell characteristics of the two new engineered viruses (AAT and shRNA) were evaluated and compared with the HMGA1 sequestration virus (HBS) in three different human pancreatic cancer cell lines (primary focus) and one human breast cancer cell line (to test generality with one non pancreatic cancer cell line) using *in vitro* viability, toxicity and necrosis assays. The

effect of treatment on cancer cell migration capability was evaluated using classic wound healing assays. The effect of each treatment on anchorage-independent growth capacity and growth inhibition was evaluated using soft agar proliferation assays. The numbers of HMGA1 mRNA transcripts per cell were quantified (i.e., copy numbers per cell determined) using qPCR and quantification of the cell numbers from which total RNA was isolated. Effects on HMGA1 protein levels were evaluated using quantitative western blot experiments. A putative cis-acting HMGA1 natural antisense transcript (NAT) was experimentally established and found to be expressed both in normal healthy pancreatic and breast epithelial cells and in pancreatic and breast cancer cells. qPCR was used to compare relative HMGA1 NAT levels between human epithelial and cancer cells. Finally, the effects on HMGA1 NAT levels were quantified and correlations between NAT levels and phenotypic effects of cancer cells were analyzed, all of which is described and discussed below.

## Materials and methods

### Construction, purification, and quantification of engineered adenovirus vectors

Engineered adenoviruses were developed using the OD260 Inc system by combining four plasmids (pAd1127−07 (https://od260.com/product/pad1127-07), pAd1128 (https://od260.com/product/pad1128), pAd1129 (https://od260.com/product/pad1129), pAd1130 (https://od260.com/product/pad1130)) to assemble the whole adenoviral genome into a single cosmid (S Fig. 1A in S1 File). The HMGA1 antisense sequence (S Fig. 1B in S1 File) was amplified from an *E. coli* vector and the shRNA sequence was obtained from the RNA consortium (S Fig. 1C in S1 File) and synthesized commercially. Target sequences were inserted into the pAd1127 plasmid multiple cloning site under control of the cytomegalovirus (CMV) promoter (S Fig. 2 in S1 File) and confirmed by sequencing following PCR amplification (S Fig. 3A in S1 File). Finally, all four plasmids were ligated to generate a cosmid containing the intact engineered adenovirus vector. Successful cosmid generation was confirmed by evaluating characteristic plasmid fragment patterns expected following enzymatic digestion, as provided and confirmed by the manufacturer (S Fig. 3B-C3 column in S1 File). The cosmid was linearized and transfected into AD293 cells to generate the engineered viruses for subsequent viral DNA infection experiments and plaque assays were conducted to confirm virus production (S Fig. 4 in S1 File). Viruses generated for these studies were replication defective. The following viruses were produced and used in the study: control adenovirus (CV) serotype 5 (No modification to the parent genome), artificial antisense transcript (AAT) virus designed to express the antisense complement of the mRNA HMGA1 sequence under the control of a CMV promoter, short hairpin RNA (shRNA) virus designed to express the shRNA sequence targeting HMGA1 under the control of a CMV promoter, and hyper binding site (HBS) virus containing HMGA1 sequestration sequence was developed and characterized previously in our lab [10]. After confirming virus generation, cells were lysed and viruses purified and quantified using plaque assays.

### Cell cultures

Human pancreatic cancer cell lines, MIA PaCa-2, BxPC-3 (RRID:CVCL_0186), PANC-I, and the human breast cancer cell line ZR-75 (RRID:CVCL_0588) were purchased from the American Type Culture Collection (ATCC). Human pancreatic epithelial ductal cells (H6c7, Kerafast) and human primary mammary epithelial cells (HMEC, from ATCC) were used as healthy control cell lines. AD293 cells (RRID:CVCL_9804) (Agilent) were used to measure effects of virus replication with replication defective viruses. Media preparation and culture maintenance were done following manufacturer's protocols. All cells were maintained at 37°C in an incubator with a 5% $CO_2$ humidified atmosphere.

### Viability assays

Cells were plated in 96-well plates at a density of 2 X $10^3$ cells per well. After 24 hours, used media was aspirated and replaced by new media and cells infected by CV, HBS, AAT and shRNA viruses at doses of 10, 50 and 100 particles per

cell (ppc). Viability was determined every 24 hours for four days. The PrestoBlue HS (Thermo Fisher) cell viability reagent was used for determining cell viability.

## LDH cytotoxicity assays

The LDH cytotoxicity assay kit (Dojindo) was used to measure membrane integrity. This assay was performed simultaneously with viability assay using the same experimental procedure as the viability assay.

## Wound healing assays

Cells were plated in a 24-well plate, allowed to grow until fully confluent, a wound made through the cell monolayer (0 hour) and virus added to each well. Wounds were monitored at 24 and 48 hours, images taken for wound measurement and analyzed using the ImageJ software (RRID:SCR_003070).

## Soft agar cell proliferation assays

Soft agar proliferation assays were performed as follows. Cells were treated with different viruses, cells mixed with agar solution and the suspension plated to each well. Cell numbers were calculated before pouring cells into six well plates. Cells were allowed to grow until visible colonies were formed. A 1% NBT (nitro-blue tetrazolium chloride) staining solution (in PBS) was added to the wells and incubated overnight at 37°C to visualize colonies, after which images were collected and colonies quantified using the ImageJ software (RRID:SCR_003070).

## Apoptosis and necrosis assays

The RealTime-Glo Annexin V Apoptosis and Necrosis Assay kit (Promega) was used by plating 2 X 103 cells per well in 96 well plates and the cells treated with viruses after 24 hours. Both apoptosis (luminescence) and necrosis (fluorescence) were measured every 12 hours for up to six days (144 hours) using a microplate reader.

## Western Blot analyses

Cells were infected with engineered viruses and nuclear proteins extracted (nuclear and cytoplasmic extraction kit, BosterBio) and quantified using a bicinchoninic acid assay. Proteins were separated by SDS-PAGE, transferred to PVDF membrane followed by blocking in 5% non-fat milk powder solution. The membrane was then hybridized with primary antibody (anti-TATA (ab220788) for control protein loading and anti-HMGA (ab129153) as target protein) followed by secondary horseradish peroxidase antibody in tris-buffered saline with tween-20 (TBST). Proteins were detected by using an ECL plus detection kit (BosterBio), images taken by ChemidocMP (Bio-Rad), and proteins quantified and normalized by ImageLab software (Bio-Rad). The quantitative Western blot analyses were performed by using a densitometer to normalize the amount of HMGA1 detected relative to the quantified and normalized amount of TATA transcription factor housekeeping gene product loaded.

## RT-PCR and qPCR analysis

Total RNA was extracted (RNeasy Plus mini kit, Qiagen) from virus infected cells and quantified followed by reverse transcription (Quantitect, Qiagen) and qPCR (Multiplex PCR kit, Qiagen). A multiplex qPCR instrument (Rotor gene Q, Qiagen) was used for detecting and quantifying (Q-Rex software, Qiagen) HMGA1 transcript copy numbers. All primers for HMGA1 mRNA, antisense transcript from viral genome and detection of NAT transcripts are in S Table 1-2 in S1 File and the optimization and best PCR conditions for NAT primers are shown in S Fig. 5. in S1 File qPCR amplified target sequences, synthesized and purchased from Integrated DNA Technologies (IDT), were used as standard templates for creating calibration curves to calculate copy number of all targeted transcripts per cell (S Fig. 6 in S1 File). All qPCR primer products were sequence verified and validated following the MIQE guidelines [13].

### Statistical analyses

All statistical data analysis and data visualization were performed in the R programming language (RStudio-2024, version 4.3.3). The details for all statistical testing including model selection and model validation are described in S Method 1 in S1 File.

## Results and discussion

### Cells treated with the HMGA1 sequestration (HBS) virus showed significantly reduced viability compared to treatment with the AAT and shRNA viruses

Viability assays were conducted with MiaPaCa-2, PANC-1 and BxPC-3 pancreatic cancer cell lines and with the ZR-75 breast cancer cell line, in addition to two control cell lines (healthy pancreatic epithelial cells (E6c7) and healthy breast epithelial cells (HMEC)). Raw viability assay data collected for MiaPaCa-2 cells infected with viruses at a dose of 50 particles per cell (ppc) is shown in **Fig. 1A** indicating a significant drop in viability with HBS virus. Infection with the CV showed minimal effect and decreasing cell viability was observed with the shRNA virus, AAT virus and HBS virus, respectively. This trend was generally replicated in all cell lines, as can be seen from quantified data plotted at 72-hours post infection in **Fig. 1E**. Decreased cell viabilities observed with AAT (~25–50% reduction) and HBS (~50–75% reduction) viruses were statistically significant, whereas effects with the shRNA virus (~25–35% reduction) were generally not statistically significant despite a trend of decreased viability. Viability data for all cell lines from day 1 to day 4 under various treatments (10 ppc, 50 ppc, 100 ppc) are summarized in S Fig. 7-14 in S1 File.

### Cancer cell migration capacity in wound healing assays mirrored cell viability results

Cancer metastasis requires detachment of cancer cells from primary tumors and migration to a new location remote from the original tumor site [14, 15]. Wound healing assays enable quantitation of collective cancer cell migration capability in response to therapeutic treatment. Wound healing assays using BxPC-3 cells are shown in **Fig 1B**. At 24 and 48 hours post wound, HBS virus treated cells were significantly delayed in wound healing compared to CV infected and non-infected (negative control, NC) cells. Quantitative data for all wound healing experiments after 48 hours is summarized **Fig. 1E**, showing that the HBS virus treatment had a larger effect (60–70% reductions) compared to the AAT and shRNA viruses (25–50% reductions), however all three viruses caused statistically significant reduction in wound healing rates (**Fig. 1B, inset**). Statistical analyses for all wound healing assay data at 24 hours post wounds are shown in S Fig. 15 in S1 File. Wound healing results for all cell lines and virus treatments at 48 hours closely mirrored cell viability profiles (**Fig. 1E**).

### Treatment with all engineered viruses caused significant reduction in anchorage-independent migration capacity based on soft agar cell proliferation assays

Soft agar proliferation assays were used to evaluate the effect of virus treatments on cancer cell anchorage-independent migration potential. The soft agar proliferation assay prevents attachment of cancer cells to an extracellular matrix, which is a requirement of normal cells to grow and divide. In contrast, transformed cancer cells have the capability for anchorage-independent growth [14, 15], which facilitates cancer cell movement and formation of visible colonies in soft agar experiments. **Fig 1C** shows raw proliferation data for ZR-75 cells. Non-infected cells served as a negative control (NC) (**Fig 1Ci**). CV treated cells caused no significant reduction in proliferation (**Fig 1Cii**) in comparison to the non-infected NC control (**Fig 1Ci**). Strong reductions in cell proliferation (40–50%) were observed following treatment with the HBS, AAT and shRNA viruses (**Fig 1Ciii-1Cv, respectively**). Larger reductions in proliferation were observed for ZR-75 breast cancer cells and BX-PC-3 pancreatic cancer cells, which retained only 30–40% of untreated proliferation capability (**Fig 1E**). Raw data for all proliferation assays were shown in S Fig 16A-C in S1 File.

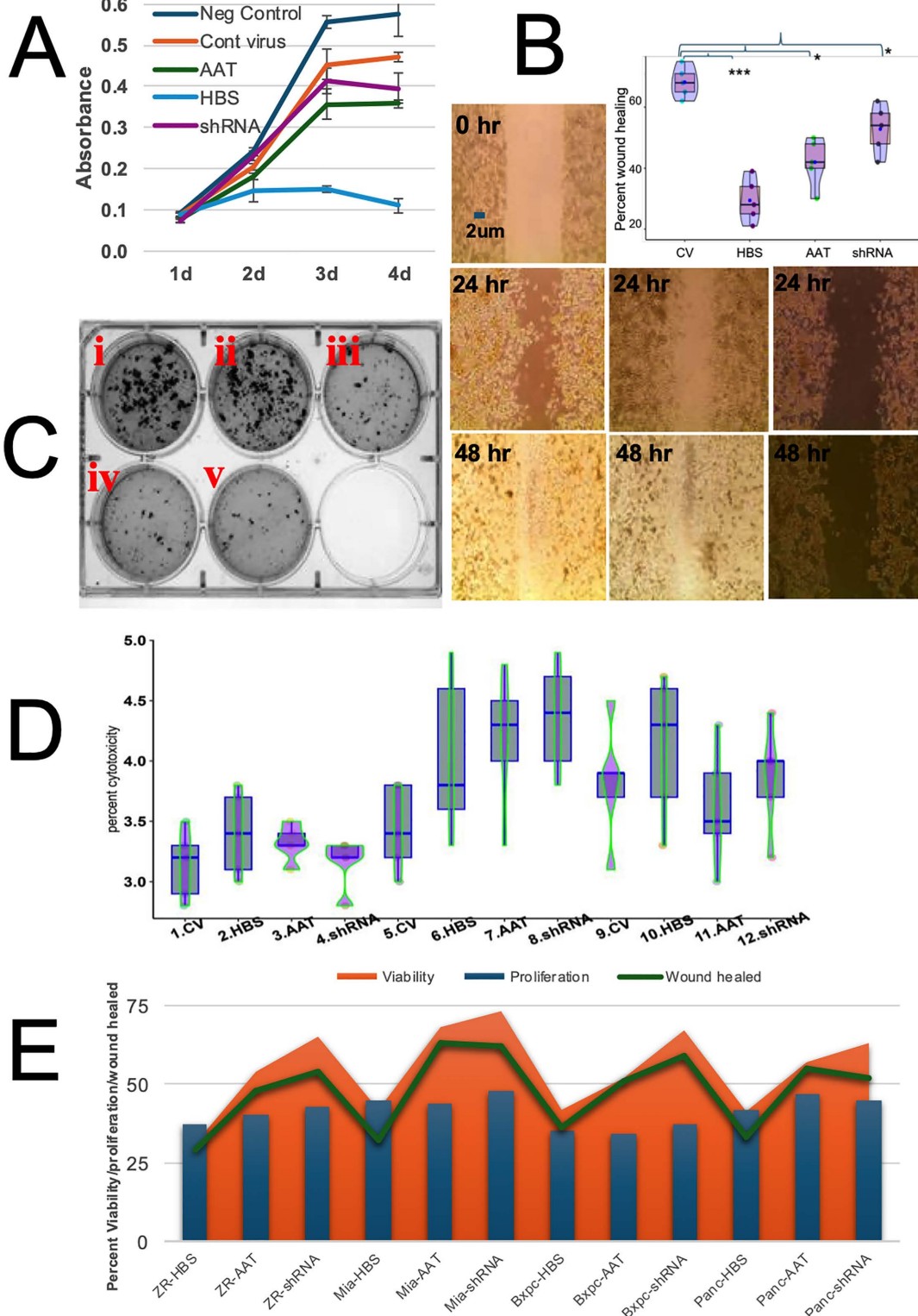

**Fig 1.** ***In vitro*** **assays for engineered adenovirus infected human pancreatic and breast cancer cell lines. A)** Viability assay of MiaPaCa-2 from day 1 to day 4 following 50 ppc infection. HBS virus infected cells had significantly less viability (p < 0.001) compared to other virus infected cells. **B)** Raw and quantified wound healing assay data for BxPC-3 cells. The first-row image shows an initial wound at 0 hour. The second and third row images show

wound healing of cells after 24 and 48 hours respectively. Left, middle and right column images indicate no virus infection (NC), control virus infected cells (CV), and HBS virus infected cells, respectively. The inset graph shows a plot of the percent wound healed for each virus relative to the CV with statistically significant differences connected by horizontal lines (ANOVA followed by posthoc analysis). C) Soft-agar cell proliferation assay of ZR-75. Each well indicates a different condition as follows: (i) no virus infected cells (NC), (ii) CV infected cell, (iii) HBS infected cells, (iv) AAT infected cells, (v) shRNA infected cells. D) Cytotoxicity assays for ZR-75 cells on day 4 at different viral treatments: 10 ppc (1-4), 50 ppc (5-8), 100 ppc (9-12). E) A combined plot of viability, proliferation and wound healing assay for all engineered viral infections of ZR-75, MiaPaCa-2, BxPC-3 and PANC-1 cells. Detailed statistical calculations of each of these experiments were shown in supplementary figures (all viability assays in S Fig. 7-14 in S1 File, wound healing assay in S Fig. 15 in S1 File and proliferation assay in S Fig. 16 A-C in S1 File).

## Infection with the control and engineered viruses caused no cytotoxic effects

Cytotoxicity assays were performed to determine if infection with engineered viruses caused cell toxicity due to virus infection itself, independent of the engineered HMGA1 targeting elements carried by modified viral vectors. The lactate dehydrogenase (LDH) cytotoxicity assay measured cell membrane disruption following viral infection. The 3–5% percent cytotoxicity observed, as illustrated for the ZR-75 cell line in **Fig 1D**, indicated no significant cytotoxicity due to loss of cell membrane integrity for any of the cell lines, which were measured from day 1 to day 4 following viral infections (S Fig 17-20 in S1 File).

## Virus-mediated cell death was caused by early apoptosis followed secondary necrosis

Combined apoptosis and necrosis assays were performed to determine the mechanism of virus-induced cell death. The combined annexin V luciferase-based apoptosis assay measured loss of cell membrane integrity based on detection of exposed phosphatidylserine on the outside of the cell wall during early apoptosis, and a DNA binding dye that detects necrosis due to cell wall rupture and exposure to the genomic DNA of the necrotic cells. All cancer cells treated with engineered viruses exhibited apoptosis that peaked from 24–48 hours post infection (**Figs 2A**, **2C**, **2E and 2G**) and ended by ~36 hours post infection. CV infection did not show apoptosis in any of the cell lines. The apoptosis responses varied in the different cell lines. The MiaPaCa-2 cells responded about the same to all three viruses (**Fig 2C**) whereas ZR-75 and BxPC-3 cells were most sensitive to the HBS virus (**Figs 2A and 2E**). All cell lines exhibited secondary necrosis that lagged about four days after peak apoptosis (**Figs 2B**, **2D**, **2F and 2H**). These data supported the cytotoxicity assays illustrating that infected cells remained viable during early apoptosis until secondary necrosis occurred four days later. While there is evidence for triggering of apoptosis by all three engineered viruses, the molecular details for this induction are not known. All three viruses cause reduction in the cellular levels of HMGA1. Therefore, one can consider the potential downstream effects that lead to increased apoptosis upon reduction of HMGA1 levels. For example, HMGA1 is known to inhibit the p53 tumor suppressor protein [16, 17], thus blocking its ability to trigger apoptosis, therefore reductions in HMGA1 levels could stimulate increased apoptosis. Furthermore, HMGA1 has been shown to interact directly with p53 in a manner that blocks binding to the Bax promoter, which is a p53 and p21waf1 effectors, further blocking the tumor suppressor function of p53 [18]. It has also been reported that elevated levels of HMGA1 cause upregulation of Il-15 and Il-15Ra, which causes anti-apoptotic effects by inducing expression of apoptosis inhibitors Bcl2/Bcl1/Bcl-x(L) [19–21].

## Cellular HMGA1 protein levels decreased in virus infected cells

Whereas the HBS virus was designed to sequester overexpressed HMGA1 at decoy hyper binding sites, the AAT and shRNA viruses were designed to suppress translation of HMGA1 mRNA into HMGA1 protein. Therefore, reduced HMGA1 protein levels were expected in the AAT and shRNA virus infected cells. Quantitative Western blot analyses (S Fig. 21, S Fig. 22 in S1 File) performed on BxPC-3 cells at a dose of 50 ppc (**Fig 3A**) indicated that treatment with the HBS virus reduced HMGA1 protein to less than 40% compared to untreated cells. Treatment with AAT and shRNA viruses reduced

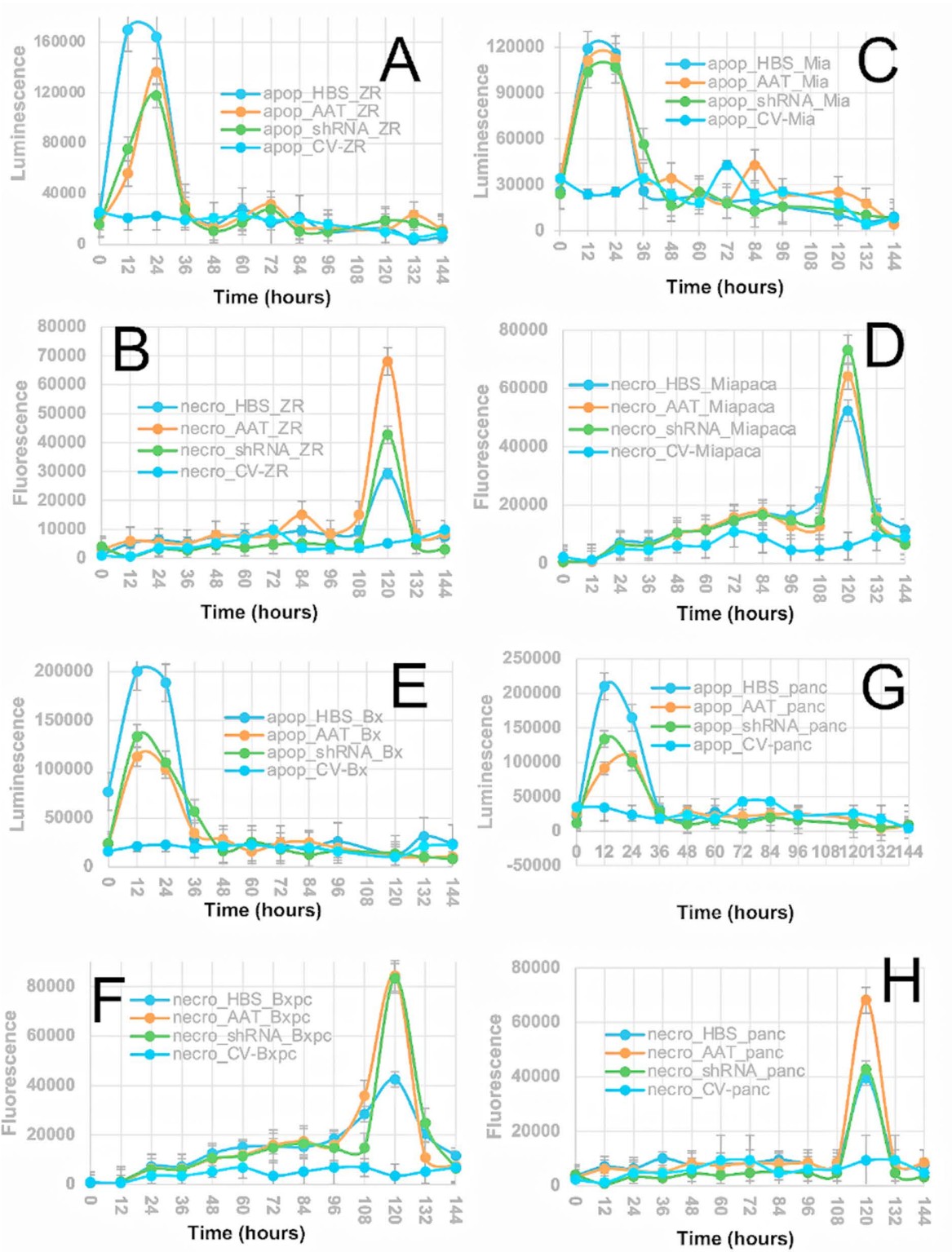

**Fig 2. Apoptosis and necrosis assays for virus infected cells. A, C, E, G)** Luminescence-based detection of early apoptosis for ZR-75, MiaPaCa-2, BxPC-3 and PANC-1 cells, respectively. **B, D, F, H)** Fluorescence-based detection of necrosis for ZR-75, MiaPaCa-2, BxPC-3 and PANC-1 cells, respectively. The virus used for infection in each experiment is indicated in the legend for each panel, e.g., apop_HBS_ZR indicates that the data represents

an apoptosis assay for ZR-75 cells following infection with the HBS virus, etc. The legends for the necrosis assays can be interpreted as follows, e.g., necro_HBS_ZR indicated that the data represented a necrosis assay following infection with the HBS virus for the ZR-75 cells, etc.

HMGA1 levels to just over 40% and about 60% relative to the untreated cells, respectively. HMGA1 protein levels following treatment with all viruses and for all cell lines are summarized in **Fig 3F**, showing that the trend observed in the BxPC-3 cells was preserved in all other cell lines.

### HMGA1 mRNA transcript levels decreased with HBS and shRNA treatment but increased with AAT virus treatment

The AAT and shRNA viruses were designed to reduce HGMA1 mRNA available for translation into HMGA1 protein. Quantitative PCR was performed to determine HMGA1 mRNA transcript copy numbers per cell to determine the effect of various virus treatments. HMGA1 mRNA transcript levels were around 300 copies per cell in untreated BxPC-3 cells (**Fig 3B**). Treatments with the HBS and shRNA viruses decreased the transcript levels to ~100 and ~200 copies per cell, respectively (**Fig 3B**). The copy number increased to over 1000 copies per cell in the AAT treated cells (**Fig 3B**), but this did not translate into increased HMGA1 protein in the cells (**Fig 3A**). Treatment with AAT virus increased HMGA1 mRNA copy numbers per cell in all cell lines (**Fig 3F**). Treatment with the HBS and shRNA viruses caused only small effects on the HMGA1 mRNA copy numbers per cell in the other cell lines (**Fig 3F**).

### A putative cis-acting HMGA1 natural antisense transcript (NAT) was detected and quantified in both in native and virus-infected cancer cells

Given that NATs regulate the expression of thousands of human genes [22–24], we searched for a putative cis-acting HMGA1 NAT and examined how its expression levels correlated with HMGA1 mRNA transcript and protein levels. Unlike coding strand pre-mRNA whose introns are removed prior to translation into proteins, NATs can contain intron regions. The HMGA1 gene contains six introns and exons (**Fig 3C**). To probe for a putative cis-acting HMGA1 NAT, potentially including introns, we used a strand-specific RT-primer walk strategy with PCR primers spanning each intron-exon junction (**Fig 3C**). Since the HMGA1 NAT mRNA would be oriented opposite to the HMGA1 coding strand, the strand-specific RT primers were designed downstream (on the putative NAT mRNA strand) of each intron-exon junction. PCR primers used to detect intron-exon junctions are listed in **S Table 1** and schematically indicated in **Fig 3C**. In the absence of genomic DNA (gDNA), a negative control (NC) experiment conducted with PCR primers in the absence of the RT enzyme would not be expected to produce PCR products. A primer walk experiment using PANC-1 cells is shown in **Fig 3D**. The NC experiment produced no bands indicating the isolated total RNA was free from gDNA contamination. A second negative control experiment was performed where RT primers were included but the PCR primers were omitted. In this case, PCR primer-independent DNA synthesis was observed for some fragments, apparently as a result of self-priming [24]. The RT-PCR experiment performed with product-specific RT primers and with random hexamer RT primers showed strong RT-PCR products for all intron-exon junctions indicating the presence of a putative cis-acting HMGA1 NAT (**Fig 3D**). Evidence for the putative cis-acting HMGA1 NAT was confirmed in all the cancer cell lines, in the AD293 cells (S Fig. 23A-D in **S1 File**) and in the two control cell lines. Hereafter, we will refer to this putative cis-acting HMGA1 NAT simply as the "HMGA1 NAT". To quantitate the HMGA1 NAT transcript levels, the three most specific and weakest self-priming intron-exon junction RT-PCR products were selected for qPCR. HMGA1 NAT copy numbers were found to be significantly lower than HMGA1 mRNA transcript levels, with most on the order of less than 10 copies per cell, but with cells treated with AAT virus increasing to around 30 copies per cell (**Fig 3E**). No significant change in HMGA1 NAT levels were observed following viral infections. **Fig 3F** provides a summary of absolute quantitation of the HMGA1 mRNA transcripts, putative HMGA1 NATs and relative HMGA1 protein levels for all cancer cells studied and for all virus treatments applied.

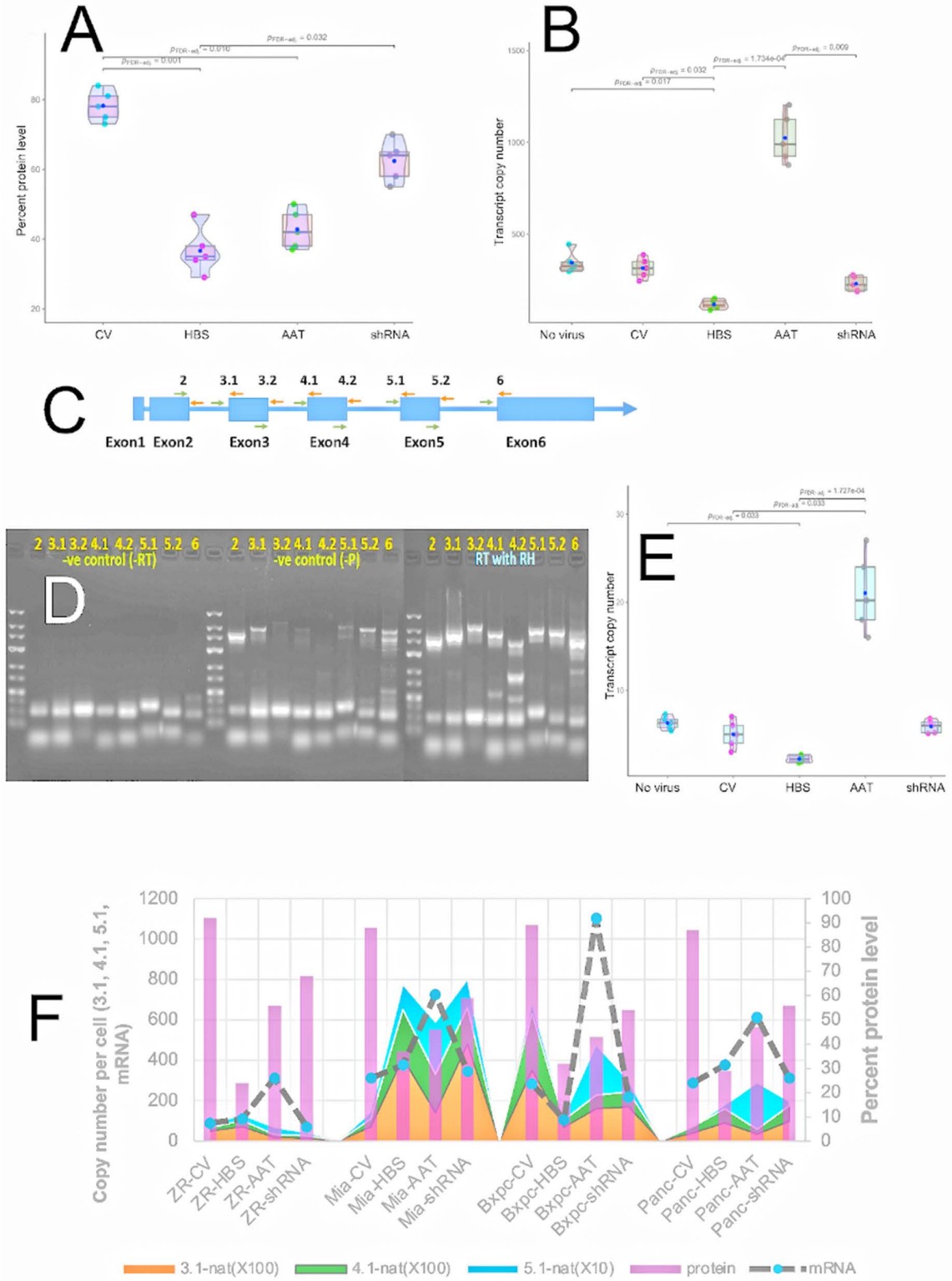

**Fig 3. Quantification of HMGA1 protein and mRNA transcript levels and the detection and quantification of cis-acting HMGA1 NAT transcripts in cancer cells. A)** Normalized HMGA1 protein levels in BxPC-3 cells following infection with different viruses indicated by the x-axis labels for cells infected at a dose of 50 ppc. Significant differences in protein levels (p < 0.05) are indicated by horizontal connected lines. **B)** HMGA1 mRNA transcript

copy numbers (per cell) for BxPC-3 cells infected with each virus at a 50 ppc dose. Significant changes in HMGA1 mRNA transcript levels (p < 0.05) are indicated by horizontal connected lines. **C)** Strategy of designing NAT primers at all possible exon-intron junctions for the sHMGA1 gene. The number and positions of exon-intron information were collected from NCBI to create the schematic diagram of the HMGA1 gene. Eight intron-exon junctions were covered from the 5' to 3' direction (left to right) for primer design. In each primer pair, one primer sat on either the intron/exon and the other primer sat on the exon/intron site. **D)** Gel analysis for detecting potential cis-HMGA1 NATs in PANC-1 cells. cDNAs produced from RT reactions. Two negative controls: No RT enzyme (left) and no RT primer (middle) are shown and the sample with all components (right)) were amplified with PCR primer pairs for all junctions. **E)** NAT transcript copy numbers (per cell) determined for all virus infections (indicated by labels on x-axis) in BxPC-3 cells at a 50 ppc dose. Significant differences in NAT levels (p < 0.05) are indicated by horizontal connected lines. **F)** A combined plot of HMGA1 protein levels (right y-axis) and HMGA1 mRNA transcript and NAT transcript levels (left y axis) for all virus infected cell lines. Since the NAT copy numbers were much lower than mRNA copy numbers, the NAT values were multiplied by 10 (for 5.1 NAT position) or 100 (for 3.1 and 4.1 NAT positions) before plotting to facilitate comparison. Detailed statistical calculation of each experiment of this combined plot is shown in S Fig. 22, 25-28 in S1 File.

## Trends between HMGA1 protein levels, HMGA1 mRNA transcript levels and HMGA1 NAT levels

The overall trends for the response of each cell line to the various engineered virus treatments were evaluated and a summarized in Table 1. First considering the effect of the HBS virus, the HMGA1 mRNA levels were almost unaffected in the ZR-75, and varying levels were observed in the different pancreatic cancer cell lines, however, a significant drop was observed in the AD-293 cells. At the same time, the HMGA1 NAT levels appeared only weakly affected, with a slight increase in the ZR-75 cells, a small increase in the pancreatic cancer cells (except BxPC-3) and a relatively large decrease in the AD-293 cells. However, large reductions in HMGA1 protein levels were observed in all cell lines, with a 72% drop in HMGA1 protein levels in ZR-75 cells compared to control cells, an average 59% drop of HMGA1 protein in pancreatic cancer cells compared to the controls and the largest drop in HMGA1 protein levels observed for AD-293 cells (78%). The significant drops in HMGA1 protein levels without a significant reduction in HMGA1 mRNA protein levels indicated an, as of yet, uncharacterized negative feedback mechanism. The molecular mechanism or pathway response of such a negative feedback mechanism could be probed with additional experiments, e.g., transcriptomic or proteomic analysis of downstream targets to elucidate how HMGA1 sequestration contribute to auto-down-regulation. In contrast, treatment with the shRNA virus caused a reduction in the copy number of the HMGA1 mRNA transcripts (except MiaPaCa-2), as expected, with a corresponding drop in the HMGA1 NAT levels, and a corresponding drop in HMGA1 protein levels (HBS$_{(average)}$: 56.8%, ZR-75: 62%, AD-293: 55%) compared to the drop in HMGA1 mRNA levels. The result after treatment with the AAT virus was rather surprising in that the amount of HMGA1 mRNA levels increased nearly three-fold in all cell lines, with a commensurate increase in the HMGA1 NAT levels, but the HMGA1 protein levels decreased (HBS$_{(average)}$: 43.3%, ZR-75: 61%, AD-293: 55%) by roughly the same amounts as observed for treatment with the shRNA viruses. This result was consistent with the HMGA1 mRNA transcripts being protected from degradation when hybridized with the HMGA AATs, but not being available for translation, thus leading to reduced HMGA1 levels. Direct evidence for such an interaction could be obtained experimentally, for example, by using RNA immunoprecipitation to detect AAT-mRNA hybrids.

## Regression analysis of the relative contributions of cell growth inhibition and cell death to decreases in cell proliferation

HBS-treated cells behaved differently than AAT- and shRNA-treated cells in that AAT- and shRNA-treated cells exhibited higher cell viabilities (50–75% relative to the CV) compared to proliferation capabilities (40% − 50% compared to the CV) for all cell types whereas the viabilities and proliferation capabilities were very similar for the HBS-treated cells for all cell types (30–50%) **(Fig 1E)**. To understand this different behavior, the relative contributions of cell growth inhibition and cell death to reduced cell proliferation and reduced cell viability were assessed using a modified lactose dehydrogenase (LDH) cytotoxicity assay [25], which enabled quantification of both dead cells (cell death) and growth inhibition in cells that were still alive (viable

Table 1.  Summary of HMGA1 mRNA transcript levels, HMGA1 NAT levels and HMGA1 protein levels in the pancreatic and breast cancer cell lines following treatment with the HBS, shRNA and AAT viruses. Values in parenthesis indicate standard deviations calculated from three technical replicates for each of three or more biological replicates.

| | | No Infection | | | HBS Infected | |
|---|---|---|---|---|---|---|
| | ZR-75 | MiaPaCa-2/BxPC3/Panc-1 | AD-293 | ZR-75 | MiaPaCa-2/BxPC3/Panc-1 | AD-293 |
| mRNA transcripts (copies/cell) | 92 +/- 26.1 | MiaPaCa-2: 312.4+/-13.4 | 370 +/-78.4 | 102 +/-16.1 | MiaPaCa-2: 381.4+/- 24.4 | 112+/-25.5 |
| | | BxPC-3: 286.5+/- 20.4 | | | BxPC-3:156.5+/- 17.7 | |
| | | Panc-1: 293.4+/-19.4 | | | Panc-1: 354.4+/- 31.9 | |
| NAT transcripts (copies/cell) | 2.4 +/- 0.78 | MiaPaCa-2:3.7+/- 1.1 | 9.5+/- 1.2 | 3.6 +/- 0.88 | MiaPaCa-2: 5.7+/- 4.1 | 2.6 +/- 0.65 |
| | | BxPC-3: 9.5+/- 2.4 | | | BxPC-3: 2.5+/- 1.4 | |
| | | Panc-1: 1.9+/- 0.34 | | | Panc-1: 2.6+/- 1.1 | |
| HMGA1 protein | 100 | MiaPaCa-2: 100 | 100 | 28 | MiaPaCa-2: 47.3+/- 7.1 | 22 |
| | | BxPC-3: 100 | | | BxPC-3: 36.5+/- 4.4 | |
| | | Panc-1: 100 | | | Panc-1: 39.3+/- 5.1 | |
| | | AAT Infected | | | shRNA Infected | |
| | ZR-75 | MiaPaCa-2/BxPC3/Panc-1 | AD-293 | ZR-75 | MiaPaCa-2/BxPC3/Panc-1 | AD-293 |
| mRNA transcripts (copies/cell) | 276 +/-50.5 | MiaPaCa-2: 715.4+/- 38.4 | 1000 +/-303.8 | 60 +/- 9.8 | MiaPaCa-2: 356.4+/- 18.4 | 240 +/-53.4 |
| | | BxPC-3: 1102.2+/- 48.3 | | | BxPC-3 238+/- 18.3 | |
| | | Panc-1: 592.9+/- 44.5 | | | Panc-1: 264.4+/- 24.5 | |
| NAT transcripts (copies/cell) | 3.7 +/- 0.92 | MiaPaCa-2: 7.8+/- 2.1 | 24.1+/- 5.5 | 1.3+/- 0.48 | MiaPaCa-2: 4.7+/- 2.1 | 6.2 +/- 1.3 |
| | | BxPC-3: 10.5+/- 1.4 | | | BxPC-3: 1.5+/- 1.4 | |
| | | Panc-1: 5.2+/- 1.1 | | | Panc-1: 1.2+/- 1.1 | |
| HMGA1 protein | 61 | MiaPaCa-2: 49.2+/- 8.1 | 55 | 62 | MiaPaCa-2: 59.2+/- 11.1 | 55 |
| | | BxPC-3: 44.3+/- 9.4 | | | BxPC-3: 54.5+/- 7.4 | |
| | | Panc-1: 36.3+/- 10.8 | | | Panc-1: 56.7+/- 8.1 | |

cells). A regression analysis was performed to determine to what extent decreases in cell proliferation were due to increased cell death or to increased cell growth inhibition in still viable cells. A quantile regression was performed as the data was non-parametric and therefore there was no need to make any assumptions due to lower sensitivity to outlier effects [26]. In **Fig 4A**, no statistically significant cell death or growth inhibition was observed for CV infected cells. As shown in **Fig 4B**, cell death was the main cause of reduced cell proliferation in HBS infected cells ($p < 0.001$) with cell death being a statistically significant contributor to reduction in cell proliferation in all three quantiles. In contrast, only cell growth inhibition (cells still viable, but not able to proliferate) was a statistically significant contributor to reduced cell proliferation in AAT and shRNA virus infected cells in the q10 or q50 quantiles, and cell death was not a statistically significant contributor to reduced cell proliferation (**Fig 4C-4D**). This analysis explained the higher viability compared to proliferation capability observed in AAT and shRNA infected cells, i.e., they had higher numbers of viable cells, but lower proliferation was observed because fewer colonies could form due to greater growth inhibition than in the HBS infected cells (which had significantly higher numbers of dead cells).

The overall patterns from the regression analysis are summarized in the heat plot in **Fig 4E**. For example, cells treated with the CV exhibited high cell proliferation (values greater than 0.75 shaded in dark pink in the 4th column) with no significant reduction in cell proliferation due to cell death or cell growth inhibition (low effect sizes indicated by sky blue color in columns 1−3). In contrast, cells treated with HBS viruses experienced the largest decreases in proliferation (cell proliferation values near 0 shaded in sky blue in column 4) due to both cell death and cell growth inhibition (effect sizes with values near 1 in columns 1 & 3, respectively) in comparison to the AAT and shRNA treated cells, which exhibited slightly higher cell proliferation (0.25–0.5 in column 4) and higher cell growth inhibition effect sizes (0.5–0.75 in column 3)

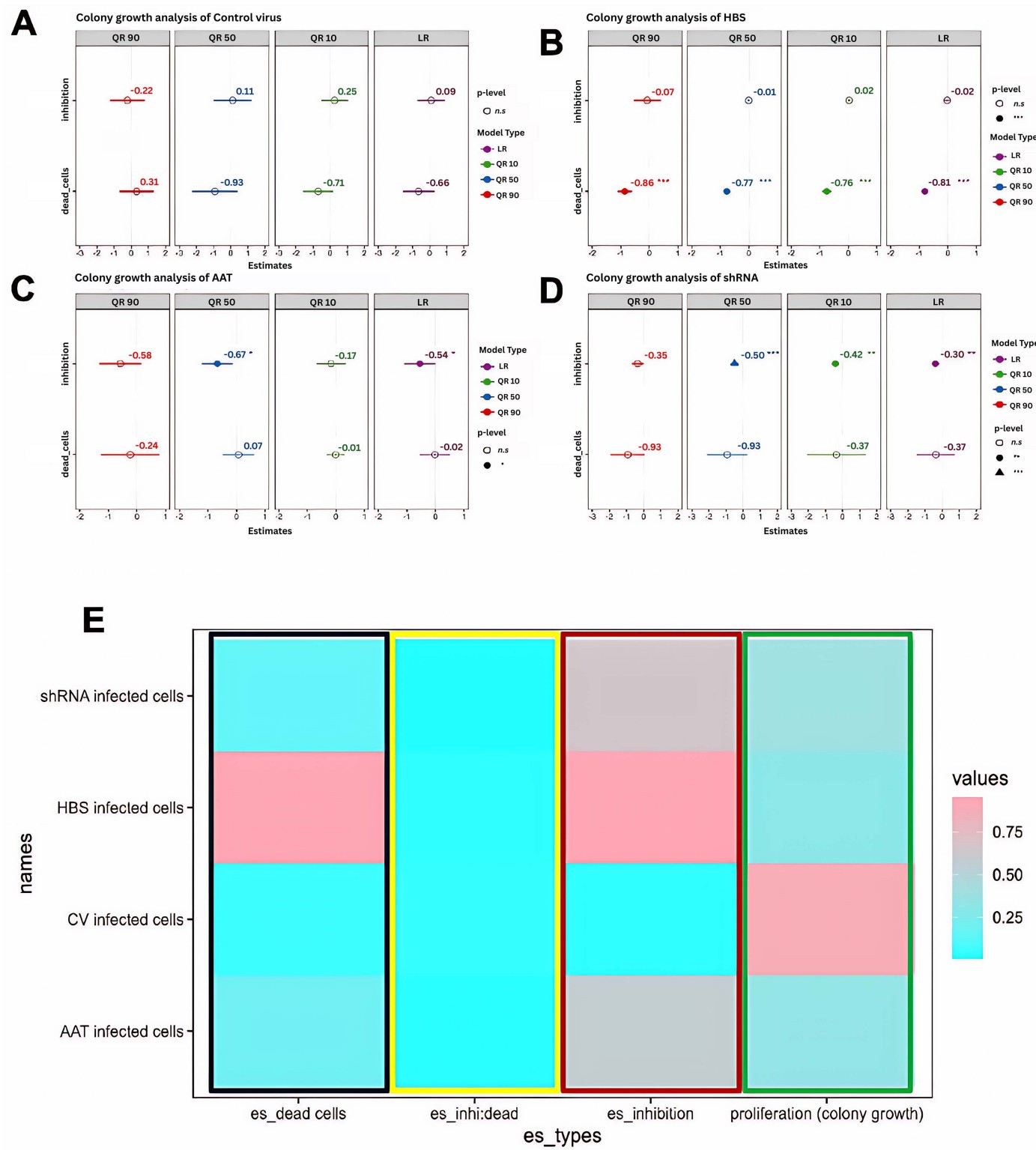

**Fig 4. Quantile regression analysis of cell inhibition and cell death effects in determining cell proliferation. A)** Control virus infected cells showed no effect of inhibition/death on colony formation as cells remained healthy all the way of the experiment. **B)** HBS infected cells showed a significant effect of cell death on colony formation. It showed that reduced colony number was mostly due to cell death. **C)** AAT virus infected cells showed

some effect of inhibition on colony formation. It indicated that cells were not dead, rather they were unable to multiply to make colonies. D) shRNA virus infected cells showed significant effect of inhibition on colony formation. It indicated that cells were not dead, rather they were unable to multiply to make colonies. The 10% quantile (q10), 50% quantile (q50), 90% quantile (q90) and linear regression (lr) data are shown in A-D with the p-values as such $p < 0.05(*)$, $p < 0.01(**)$, $p < 0.001(***)$. **E)** Heat map plot of effect sizes of dead cells (es_dead cells, black box), inhibition (es_inhibition, red box) and interaction between dead cells and cell inhibition (es_inhi:dead, yellow box) in contributing overall cell proliferation level (green box).

but relatively small effect sizes due to cell death (0.25–0.4 in column 1). Therefore, the AAT and shRNA viruses reduced cell proliferation primarily due to cell growth inhibition whereas the HBS viruses caused decreased proliferation due to a combination of both cell death and cell growth inhibition.

### The role of NATs in regulating HMGA1 transcript stability

Previous papers have reported that NATs stabilize mRNAs through direct interaction in mRNA-NAT duplexes. Formation of such duplexes regulate mRNA stability in the cytoplasm and potentially block microRNA binding to mRNA sequences, therefore preventing mRNA degradation [27, 28]. Based on this insight, we investigated whether HMGA1 transcripts and NAT levels showed any statistical correlations. For breast epithelial cells, the mRNA transcript copy number was around 6/cell and the NAT copy numbers at all positions were about 0.01/cell (S Fig. 24A-D in S1 File). For non-infected ZR-75 cells, mRNA transcript copy numbers were about 90/cell (S Fig. 25A in S1 File) and NAT copy number at 3.1, 4.1, 5.1 positions were about 0.26/cell, 0.16/cell, 2.1/cell, respectively (S Fig. 26A, 27A, 28A in S1 File). The mRNA levels increased by approximately 15x in breast cancer cells compared to in healthy cells. NAT levels at positions 3.1 and 4.1 increased by 10-20x and at position 5.1 was approximately 200x higher than in healthy cells. Similarly for pancreatic epithelial cells, mRNA transcript copy number were around 3.1/cell and NAT copy number at 3.1, 4.1, 5.1 positions were about 0.07/cell, 0.042/cell, 0.05/cell, respectively (S Fig. 24A-D in S1 File). For non-infected MiaPaCa-2, BxPC-3 and PANC-1 cells, the average mRNA transcript copy number was about 300–400/cell (S Fig. 25B-D in S1 File) and NAT copy number at 3.1, 4.1, 5.1 positions were about 1.9/cell, 0.8/cell, 11.2/cell, respectively (S Fig. 26B-D, 27B-D, 28B-D in S1 File). Therefore, in pancreatic cancer cells, the mRNA levels increased by approximately 100-130x compared to in healthy cells. Finally, the NAT level at 3.1 and 4.1 position increased by 20-30x and at 5.1 position, it increased by 220-230x compared to in healthy cells. These data illustrated that the increase of mRNA was positively correlated to increased NAT levels in a range of 10–100 fold. Therefore, cis-NATs may stabilize HMGA1 mRNA transcripts through cytoplasmic interaction of the NAT with the complementary single-stranded region of the mRNA sequence through interactions of the network of mRNAs, AS transcripts, micro-RNAs and RNA-binding proteins [27]. Another study has shown that mRNA stabilization by NATs can be mediated by cytoplasmic interaction of NATs with single-stranded regions of mRNAs that blocks microRNA destabilization of the mRNA [27, 28]. HMGA1 knock-down or overexpression experiments could be conducted to clarify the mechanism by which HMGA1 NATs stabilize the HMGA1 mRNA or modulate its translation.

### Relative contributions of HMGA1 mRNA and HMGA1 NATs in determining HMGA1 protein levels in normal healthy human breast and pancreatic epithelial cells, human breast and pancreatic cancer cells and human embryonic kidney AD 293 cells.

HMGA1 mRNA transcript levels, HMGA1 NAT levels and HMGA1 protein levels were absolutely quantified (i.e., copy numbers per cell determined) for all cell types and viral treatments. A multilinear regression analysis was performed to determine interdependency of HMGA1 mRNA transcript and HMGA1 NAT levels in determining HMGA1 protein levels and how these interactions changed with various viral treatments (S Fig. 29-50, S table 10-17 in S1 File). Regression data for all cell lines without viral infections was analyzed as a baseline (Fig 5A and S Fig. 29-34 , S table 10, 12 in S1 File). Healthy

breast epithelial cells and pancreatic epithelial cells exhibited very low HMGA1 protein levels near 0 (**Fig 5A**, **4th column**) with the highest effect sizes observed for contributions of mRNA, NATs and their interactions regulating HMGA1 levels (**Fig 5A**, **columns 1–3**). All pancreatic cancer cells without virus infections (MiaPaCa-2, BxPC-3 and PANC-1 cells) had high HMGA1 protein levels (**Fig 5A**, **4th column**) but MiaPaCa-2 cells had lower effect sizes (0.5–0.75) for mRNA and PANC-1 cells had lower effect sizes (0.4–0.6) for NATs, whereas all three cell lines had low interaction effect sizes (~ 0.5) indicating weak interactions between mRNA and NATs in regulating HMGA1 levels (**Fig 5A**, **columns 1–3**). AD293 and ZR-75 cells had very high HMGA1 protein levels with low NAT effect sizes, and in addition, ZR-75 also had a very low interaction effect size indicating weak interplay between mRNA and NAT levels in determining HMGA1 protein levels (**Fig 5A**).

HBS infected cells showed the strongest anti-cancer activity in all cell lines. All HBS infected cancer cells, as well as the AD293 cells, showed low effect sizes for mRNA and NAT interactions in reducing HMGA1 levels (**Fig 5B**). PANC-1 and MiaPaCa-2 cells showed very low NAT effect sizes and ZR-75 cells showed slightly lower NAT effect sizes for determining HMGA1 protein levels (**Fig 5B** and S Fig. 35-38 in **S1 File**). This analysis indicated that HMGA1 protein levels in HBS virus infections were determined more by mRNA levels than by NAT levels, and in some cases due to a weak correlated

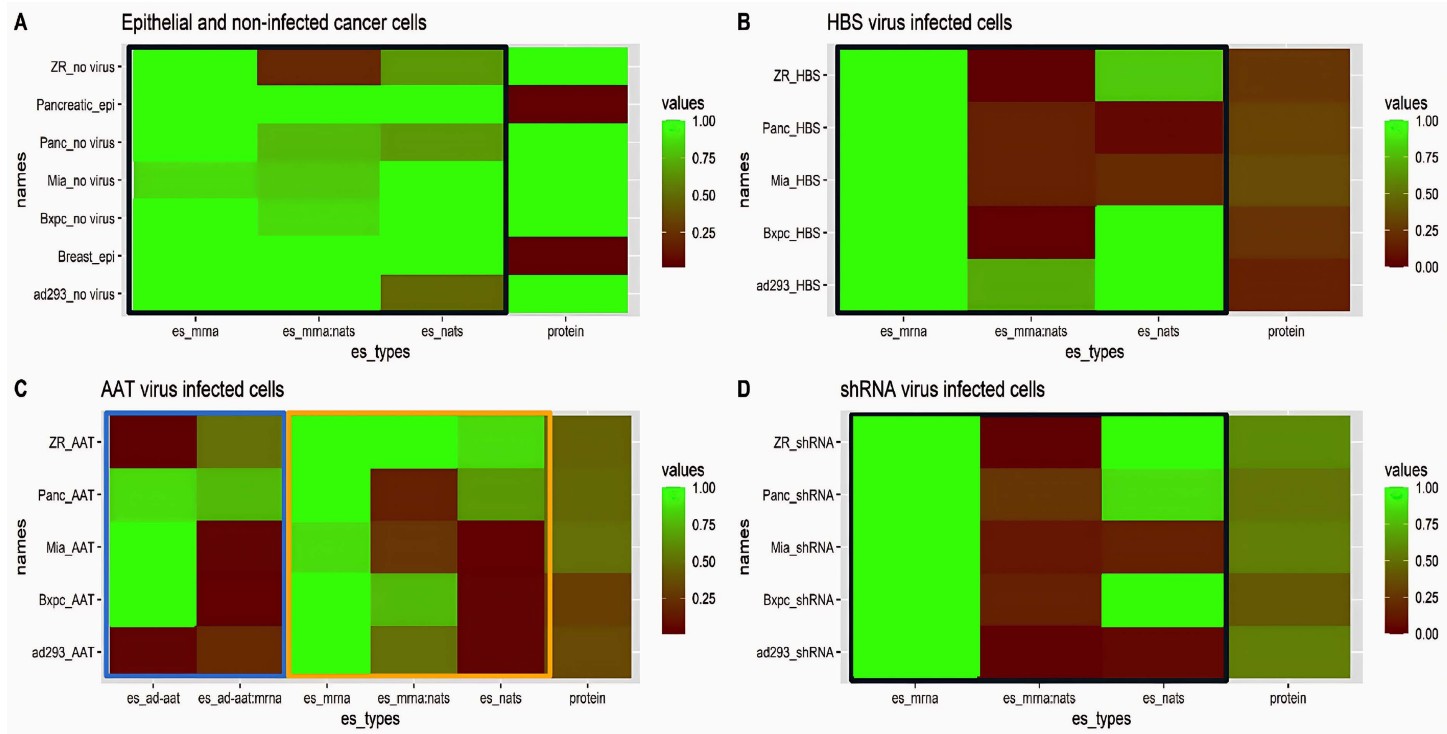

**Fig 5. Comparison of the effect of HMGA1 mRNA transcripts and NAT levels in determining HMGA1 protein levels.** In each panel, the first three columns show the effect sizes for mRNA, interaction between mRNA and NATs and NATs, respectively, and the fourth column indicates the HMGA1 protein level. The labels for the effect sizes are as follows: HMGA1 mRNA transcript (es_mrna), NAT (es_nats), their interactions (es_mrna:es_nats), AAT (es_ad-aat) and its interaction (es_ad-aat:es_mrna) are identified by the x-axis labels. The cell types and viruses used for infections are indicated by the labels on the Y axis, as follows: ZR_no virus = ZR-75 cells with no virus, Pancreatic_epi = H6c7 cells, Panc_no virus = Panc-1 cells with no virus, Mia_no virus = Mia PaCa-2 cells with no virus, Bxpc_no virus = BxPC-3 cells with not virus, Breast_epi = HMEM cells, and ad293_no virus = AD293 cells with no virus. The partial eta squared values obtained from the effect size calculations ranged from 0 to 1. The protein levels were normalized to range from 0 to 1 (0 is lowest, 1 is highest). Therefore, both independent variable and dependent variable were plotted in the same range to enable evaluation of their relationship. The exact values of the effect sizes of the independent variables were shown in S Fig. 24-50 in S1 File. In the heat map in the effect size columns, red to green color indicates lower to higher effects in determining protein levels, i.e., a low effect size indicates a lower contribution to protein production and a larger effect size indicates greater contribution to protein production. **A)** Analysis for breast and pancreatic ductal epithelial cells and non-infected cancer cells. **B)** Analysis of HBS infected cancer cells. **C)** Analysis of AAT infected cancer cells. **D)** Analysis of shRNA infected cancer cells.

interaction between mRNA-NAT compared to non-infected cancer cells as shown in **Fig 5B** (column 1 compared to columns 2 and 3).

In AAT virus infected cells, AAT levels had a high effect size in determining HMGA1 protein levels in the three pancreatic cancer cell lines, but low effect sizes for determining the HMGA1 protein levels in the breast cancer cell line and in the AD293 cell line. The NATs exhibited a weaker effect, and mRNA-NAT interactions had a greater effect in determining HMGA1 protein levels (S Fig. 39-42 in S1 File ) compared to HBS and shRNA treated cells. To probe AAT and mRNA transcript interactions, a multiple regression analysis was performed with the AAT and mRNA transcripts considered as independent variables and HMGA1 protein level considered as the dependent variable (**Fig 5C**-blue box). The interaction between AAT levels and mRNA levels had weak effect sizes for MiaPaCa-2, BxPC-3 and AD293 cells, but higher in Panc-1 and Zr-75 cells. The AAT-mRNA interaction had less effect on HMGA1 protein levels compared to mRNA-NAT interactions (**Fig 5C**-yellow box). From this analysis, it can be concluded that mRNA-AAT and mRNA-NAT transcript interactions combined to reduce overall HMGA1 protein levels in AAT treated cells. HMGA1 mRNA transcripts levels were surprisingly high in AAT infected cells compared to other virus treated cells, potentially due to AATs stabilizing the HMGA1 mRNA due to duplex formation with the AAT, as discussed above, resulting in blocked translation due steric hindrance with the translation machinery [29, 30] and consequently reduced HMGA1 protein levels. Formation of this stable duplex could also render the HMGA1 mRNA less prone to degradation resulting in the higher non-functional mRNA levels in cells that was detected in the qPCR experiments (**Fig 3F**).

The shRNA viruses were weakest in reducing HMGA1 levels, as can be seen in **Fig 5D** **and** S Fig 43−46 in S1 File. The effect size for the impact of HMGA1 mRNA levels regulating HMGA1 protein levels remained consistently high across all cell lines (**Fig 5D**). However, the effect size for the independent effect of the HMGA1 NATs was significantly weaker for MiaPaCa-2 and AD293 cells, and somewhat weaker for PANC-1 cells (**Fig 5D**). The effect size for the interdependency between mRNA and NATs was extremely weak for all cell lines with shRNA infection (**Fig 5D**).

While all of the engineered viruses were replication deficient, AD293 cells contain fragments of the adenovirus genomic DNA that complement the replication deficient viruses. Consequently, AD293 cells supported replication of the replication deficient engineered viruses in AD293 cells, and therefore, viral infection and its effects were potentially amplified in AD293 cells. In the absence of viral infection, AD293 cells displayed high HMGA1 levels (**Fig 5A**, S Fig. 47 in S1 File), as expected, as they are derived from human embryonic kidney cells, and HMGA1 levels are known to be high during embryonic development [31]. These cells displayed high effect sizes for HMGA1 mRNA in determining HMGA1 levels and low effect sizes for NATs, but the effect size for interdependency was also high (**Fig 5A**). In contrast, HBS treated AD293 cells had the lowest HMGA1 protein levels of all treated cells (**Fig 5B**), indicating amplification of the effect of the replication deficient HBS virus, and the effect sizes of HMGA1 mRNA and NATs were independently high, but their interdependence was only moderate (**Fig 5B**, S Fig. 48 in S1 File). In AD293 cells treated with the AAT virus, the regression analysis was performed to tease apart the role of HMGA1 mRNA, NATs and adenoviral AATs in regulating HMGA1 protein levels. The data indicated that the AAT viruses reduced the HMGA1 protein levels to low-to-moderate levels (0.25–0.5 of the untreated cells) in AD293 cells. A very low effect size of the adenoviral AAT was observed (**Fig 5C,** blue box), and a moderate effect size was observed for the AAT:HMGA1 mRNA interactions (**Fig 5C,** blue box). When the NATs were included in the analysis, the NATs had very low effect sizes, the mRNA:NAT interactions s had moderate effect sizes, and the HMGA1 mRNA levels had very high effect sizes (**Fig 5C, yellow box and S Fig. 49 in S1 File**), indicating that the mRNA played a dominant role in determining the HMGA1 protein levels. Finally, the HMGA1 shRNA virus reduced the HMGA1 protein levels to low-to-moderate levels (Fig 5D, S Fig. 50 in S1 File) and the regression analysis indicated that the HMGA1 mRNA effect size was very large, whereas the NATs and the interaction between NAT-mRNA levels both had very low effect sizes, indicating that the mRNA levels played a dominant role in determining HMGA1 protein levels.

 

## Conclusions

In this project, three different engineered adenovirus vectors designed to suppress oncogenic HMGA1 activity in human cancer cells were developed and tested. Experimental assays indicated that all three engineered viruses reduced cell viability, cell anchorage-independent migration and proliferation and collective cell migration capability, all while causing minimal cell toxicity, which is important when considering safety for potential downstream clinical application. Translation to clinical application will require more rigorous testing of cell proliferation and gene expression effects post-treatment to confirm safety. While reduced cancer cell characteristics were clearly associated with reduced HMGA1 protein levels, the molecular pathways affected by HMGA1 sequestration or suppressed expression have not been determined. For example, downstream pathway activation correlated to cancer cell behavior has been reported in a recent study of a related HMGB1 protein, where stimulation of exogenous HMGB1, which is ubiquitous in the tumor microenvironment and plays a vital role in tumor recurrence, was directly related to up-regulation of pathways promoting tumorigenesis and down-regulation of pathways suppressing tumorigenesis [32].

The backgrounds for the engineered viruses examined in this manuscript were adenovirus strains previously engineered to be "conditionally replicative" so that they can only replicate in cancer cells. Conditional replication originates from deletion of the 19-kDa and 55-kDa E1B genes [33–35]. The 55-kDa E1B protein inactivates the tumor suppressor protein p53, often mutated or disabled in cancer cells but functional in normal healthy cells. Therefore, adenovirus strains lacking the 55-kDa E1B gene are incapable of replicating in normal healthy cells but capable of replicating and lysing cancer cells that lack functional p53. Deletion of the 19-kDa E1B protein, which blocks downstream effects of p53 to further prevent apoptosis, provides secondary stringency for conditional replication in cancer cells only. While the engineered viral strains are capable of infecting any cells that contain a coxsackievirus and adenovirus receptor (CAR) [36], a protein in humans encoded by the CXADR gene [36], they are not able to replicate in normal healthy cells and viruses that infect healthy cells will be cleared by the immune system. This aspect of conditional replication adds an element of safety, by specific targeting of cancer cells while leaving healthy cells unaffected, which is born out in the CV data, which showed no losses in cell viability in healthy cells.

As infection with HBS virus exhibited the largest reduction of HMGA1 protein levels and caused significantly more cell death than the other viruses, we concluded that sequestration of HMGA1 protein had a higher impact on suppressing HMGA1 oncogenesis than blocking new synthesis of HMGA1. The precise mechanism by which HMGA1 sequestration reduced HMGA1 protein levels and caused increased cancer cell death remains unclear. The interpretation of the data presented here is complicated by many factors. For example, cells infected with a MOI of 50 ppc deliver 300 HMGA1 binding sites, a number that significantly outnumbers the copy numbers of the HMGA1 mRNA transcripts per cell for every cancer cell type studied. However, the number of protein molecules per mRNA transcript varies significantly by gene [37] and depends on many factors in addition to the number of mRNA transcripts per cell including mRNA stability, the speed of translation and the rate of protein degradation [38]. Another factor that must be considered is that most of the measurements were made at 72–96 hours after virus infection, which would permit at most a single doubling time (MiaPaCa-2—40 hrs, BxPC-3 48–60 hrs, Panc-1: 26–53 hrs and Zr-75: 80 hrs) under standard culture conditions. Therefore, any reduced expression due to shRNA or AAT virus infection would not block that action of the already expressed HMGA1, which would, however, be impacted by the sequestration strategy. Another factor that may have influenced the results is the relative accessibility of HMGA1 to the engineered virus HMGA1 binding sites relative to the accessibility of the genomic target HMGA1 binding sites that may be restricted in the context of the chromatin structure [39]. Further experiments may clarify these points, such as side-by-side comparison of chromatin accessibility or HMGA1-DNA binding using a ChIP-seq experiment could provide mechanistic insight into whether the HBS disrupted HMGA1's interaction with chromatin more effectively than the ATT/shRNA viruses blocked HMGA1 mRNA translation. Based on the observation that HMGA1 sequestration resulted in reduced HMGA1 protein levels, we consider the possibility that an autoregulatory negative feedback mechanism may be involved

in regulating HMGA1 levels in cancer cells. Increased cell death in cells with reduced HMGA1 levels could involve a chromatin remodeling mechanism, which HMGA1 is known to use in regulating gene expression, potentially activating apoptotic pathways that are ineffective in the presence of high HMGA1 levels. Further research is needed to determine the complex process by which HMGA1 sequestration reduces HMGA1 protein levels and causes cancer cell death.

While the *in vitro* results presented here are promising, *in vivo* experiments have not yet been performed to validate the efficacy or safety of the approach in an organismal context, which limits the translational relevance of the findings at this point. Next steps for potential translation to clinical use include *in vivo* validation, for example, the use of xenograft mouse model studies to assess the efficacy, biodistribution and toxicity in the context of a physiological study. Further obstacles that must be addressed prior to clinical use include pre-existing immunity in humans, characterization of off-target effects such as inflammatory responses and off-target gene expression effects, and sequestration to the liver and spleen and immunodominance of the vector genes over transgenes [40].

## Supporting information

**S1 File.** This file contains the supplementary methods (S Method 1), supplementary figures (S Fig. 1–51) and supplementary tables (S Tables 1–17).
(PDF)

**S1 Raw images.** **This file contains the raw gel data for all figures included in the manuscript.**
(PDF)

## Acknowledgments

MAK acknowledges support from Miami University and the Eminent Scholar Program. The authors acknowledge the efforts of several undergraduate researchers who contributed to the project including Joshua Koehler and Emily McWilliams.

## Author contributions

**Conceptualization:** Michael A. Kennedy.

**Data curation:** Michael A. Kennedy.

**Formal analysis:** Md. Sharif Hasan, Shuisong Ni, Fatema B. Kamal, Michael A. Kennedy.

**Funding acquisition:** Michael A. Kennedy.

**Investigation:** Md. Sharif Hasan, Shuisong Ni, Megan F. Blossey, Eian Vargas, Margaret B. Bogomolny, Trang Dinh, Michael A. Kennedy.

**Methodology:** Md. Sharif Hasan, Shuisong Ni, Michael A. Kennedy.

**Project administration:** Michael A. Kennedy.

**Resources:** Michael A. Kennedy.

**Software:** Michael A. Kennedy.

**Supervision:** Michael A. Kennedy.

**Validation:** Md. Sharif Hasan, Shuisong Ni, Fatema B. Kamal, Michael A. Kennedy.

**Visualization:** Md. Sharif Hasan, Michael A. Kennedy.

**Writing – original draft:** Md. Sharif Hasan, Fatema B. Kamal, Michael A. Kennedy.

**Writing – review & editing:** Md. Sharif Hasan, Fatema B. Kamal, Michael A. Kennedy.

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
