## [Decision Letter · Decision Letter 0]

29 Apr 2025

Dear Dr. Kennedy,

Thank you for submitting your manuscript to PLOS ONE. After careful consideration, we feel that it has merit but does not fully meet PLOS ONE’s publication criteria as it currently stands. Therefore, we invite you to submit a revised version of the manuscript that addresses the points raised during the review process.

We look forward to receiving your revised manuscript.

Kind regards,

Shuai Ren

Academic Editor

PLOS ONE

Additional Editor Comments:

The study is entirely in vitro (cell lines only). While the in vitro results are promising, no in vivo experiments (e.g., mouse tumor models) were performed to validate the efficacy or safety of the adenoviral treatments in an organismal context. This limits the translational relevance of the findings.

While decreased HMGA1 protein levels correlate with impaired cancer cell viability and migration, the precise molecular pathways downstream of HMGA1 sequestration or suppression are not investigated. The following related reference could be added to strengthen the manuscript: doi: 10.3389/fmed.2021.756988.

In the introduction part, the authors should emphasize the dire need of diagnosis pancreatic cancer at an early stage and following references could be added: doi: 10.4251/wjgo.v16.i4.1256; doi: 10.2147/IJN.S335588.

Although the viruses were said to have minimal cytotoxicity in healthy epithelial cells (H6c7, HMEC), more rigorous tests (such as assessing normal cell proliferation and gene expression post-treatment) would be needed to confirm safety.

The manuscript suggests that the HBS virus is "most effective" without fully acknowledging that all findings are in vitro and that translation to clinical relevance is still speculative. Strong claims about therapeutic application are premature without animal model data.

Treatment with the AAT virus increased HMGA1 mRNA levels but still decreased HMGA1 protein levels.

Reviewers' comments:

Reviewer's Responses to Questions

**Comments to the Author**

1. Is the manuscript technically sound, and do the data support the conclusions?

Reviewer #1: Yes

2. Has the statistical analysis been performed appropriately and rigorously?

Reviewer #1: Yes

3. Have the authors made all data underlying the findings in their manuscript fully available?

Reviewer #1: Yes

4. Is the manuscript presented in an intelligible fashion and written in standard English?

Reviewer #1: Yes

Reviewer #1: This study presents a novel approach to targeting HMGA1, an oncogenic protein overexpressed in various cancers, using engineered adenoviruses. The authors designed three adenoviral vectors: one to sequester HMGA1 (HBS), one to suppress its synthesis via an artificial antisense transcript (AAT), and one to degrade its mRNA via shRNA. The study demonstrates significant reductions in cancer cell viability, invasiveness, and migration, with the HBS virus showing the most pronounced effects. The work is well-designed, methodologically sound, and addresses an important gap in cancer therapeutics. However, some clarifications and additional analyses would strengthen the manuscript.

1. Mechanistic Clarity:

o The study suggests that HMGA1 sequestration by the HBS virus initiates a negative feedback mechanism, but the exact molecular pathway remains unclear. Additional experiments (e.g., transcriptomic or proteomic analysis of downstream targets) could elucidate how HMGA1 sequestration leads to apoptosis.

o For the AAT virus, the observed increase in HMGA1 mRNA levels without a corresponding rise in protein is intriguing. The authors propose mRNA stabilization via duplex formation, but direct evidence (e.g., RNA immunoprecipitation to detect AAT-mRNA hybrids) would strengthen this claim.

2. In Vivo Validation:

o While the in vitro results are compelling, the therapeutic potential of these vectors would benefit from in vivo validation, such as xenograft mouse models, to assess efficacy, biodistribution, and toxicity in a physiological context. The authors previously demonstrated safety in mice (Hassan et al., 2018), but efficacy data in tumors are lacking.

3. Comparative Analysis:

o The HBS virus outperforms AAT and shRNA viruses, but the reasons for this difference are not fully explored. Does HBS disrupt HMGA1’s interaction with chromatin more effectively than AAT/shRNA block translation? A side-by-side comparison of chromatin accessibility or HMGA1-DNA binding (e.g., ChIP-seq) could provide mechanistic insights.

4. NAT Functional Role:

o The detection of a putative HMGA1 NAT is novel, but its functional role in regulating HMGA1 expression is speculative. Knockdown or overexpression of the NAT in cancer cells could clarify whether it stabilizes HMGA1 mRNA or modulates its translation.

5. Statistical Reporting:

o Some figures (e.g., Figure 1E, 3F) summarize complex datasets but lack detailed statistical annotations. Clarify which comparisons are significant (e.g., HBS vs. AAT/shRNA) and adjust for multiple testing where applicable.

6. Introduction: To further strengthen the context of this work within current advances in gene therapy and viral vector engineering, the authors may wish to cite the following recent study in the Introduction or Discussion sections:

Nature Biotechnology (2024). 10.1038/s41587-024-02453-3

Including this reference would help readers appreciate the broader landscape of viral vector applications in oncology."

Briefly define "conditional replication" of adenoviruses for non-specialist readers.

Minor Comments:

1. Figure Clarity:

o Figure 1B: Label wound healing images with time points (0h, 24h, 48h) for clarity.

o Figure 3C-D: Include a schematic of the NAT detection strategy in the main text to aid interpretation.

2. Discussion Limitations:

o Address potential off-target effects of the adenoviral vectors (e.g., immune responses, unintended gene disruption) and how these might impact clinical translation.

3. Writing Edits:

o Abstract: Replace "~25% & 50" with "approximately 25% &50" for consistency.

**Do you want your identity to be public for this peer review?** For information about this choice, including consent withdrawal, please see our Privacy Policy

Reviewer #1: No

---

## [Author Response · Author response to Decision Letter 1]

7 May 2025

Responses to Reviewer's Comments

Comments Requiring Author Changes

1. Mechanistic Clarity:

• The study suggests that HMGA1 sequestration by the HBS virus initiates a negative feedback mechanism, but the exact molecular pathway remains unclear. Additional experiments (e.g., transcriptomic or proteomic analysis of downstream targets) could elucidate how HMGA1 sequestration leads to apoptosis.

Response – Agreed. Unfortunately, we are not in a position to conduct these additional experiments, which are outside our capabilities. However, to address this comment and to share this information with the readers, we have added the following text to the first paragraph in the “Overview of trends between HMGA1 protein levels, HMGA1 mRNA transcript levels and HMGA1 NAT levels” section:

“The molecular mechanism or pathway response for a negative feedback mechanism could be probed with additional experiments such as transcriptomic or proteomic analysis of downstream targets to elucidate how HMGA1 sequestration leads to apoptosis.”

• For the AAT virus, the observed increase in HMGA1 mRNA levels without a corresponding rise in protein is intriguing. The authors propose mRNA stabilization via duplex formation, but direct evidence (e.g., RNA immunoprecipitation to detect AAT-mRNA hybrids) would strengthen this claim.

Response – Agreed. Again, however, these additional suggested experiments are beyond our capabilities. To address this comment, and to share this information with the reader, we have added the following sentence at the end of the paragraph in the “Overview of trends between HMGA1 protein levels, HMGA1 mRNA transcript levels and HMGA1 NAT levels” section:

“Direct evidence for such an interaction could be obtained experimentally, for example, by using RNA immunoprecipitation to detect AAT-mRNA hybrids.”

2. In Vivo Validation:

• While the in vitro results are compelling, the therapeutic potential of these vectors would benefit from in vivo validation, such as xenograft mouse models, to assess efficacy, biodistribution, and toxicity in a physiological context. The authors previously demonstrated safety in mice (Hassan et al., 2018), but efficacy data in tumors are lacking.

Response – Acknowledged. However, the suggested xenograft mouse model study would represent an entirely new and distinct study that is beyond the scope of this manuscript. To capture this idea and share with the readers, we have added the following sentence to the last paragraph of the Conclusions section:

“Regarding potential translation to clinical use, the next steps would include in vivo validation, for example, the use of xenograft mouse model studies to assess the efficacy, biodistribution and toxicity in the context of a physiological study.”

3. Comparative Analysis:

• The HBS virus outperforms AAT and shRNA viruses, but the reasons for this difference are not fully explored. Does HBS disrupt HMGA1’s interaction with chromatin more effectively than AAT/shRNA block translation? A side-by-side comparison of chromatin accessibility or HMGA1-DNA binding (e.g., ChIP-seq) could provide mechanistic insights.

Response – This is a good suggestion. Again, we are not in a position to conduct these additional experiments. To convey this suggestion to the readers, we have added the following sentence to the fourth paragraph of the Conclusions section:

“A side-by-side comparison of chromatin accessibility or HMGA1-DNA binding using a ChIP-seq experiment could provide mechanistic insight into whether the HBS disrupts HMGA1’s interaction with chromatin more effectively than the ATT/shRNA viruses blocked HMGA1 mRNA translation.”

4. NAT Functional Role:

• The detection of a putative HMGA1 NAT is novel, but its functional role in regulating HMGA1 expression is speculative. Knockdown or overexpression of the NAT in cancer cells could clarify whether it stabilizes HMGA1 mRNA or modulates its translation.

Response – Agreed. However, the suggested knockdown or overexpression experiments would represent a substantial expansion of the study that we are unable to perform at this time. In order to convey the suggestion to the readers, we have added the following sentence to the end of the paragraph entitled “The Role of NATs in regulating HMGA1 transcript stability”:

“Further HMGA1 knock-down or overexpression experiments could be conducted to clarify whether the HMGA1 NATs stabilize the HMGA1 mRNA or modulate its translation.”

5. Statistical Reporting:

o Some figures (e.g., Figure 1E, 3F) summarize complex datasets but lack detailed statistical annotations. Clarify which comparisons are significant (e.g., HBS vs. AAT/shRNA) and adjust for multiple testing where applicable.

Response – The data presented in Figure 1E and Figure 3F are meant to illustrate trends in the data and facilitate comparisons across the multiple experiments, treatments and across the various cell lines tested. We are concerned that a comprehensive statistical comparison of all the data sets included in the figure would become unwieldy. Alternatively, we have removed the error bars from the plot, and instead, direct the readers to the supplementary material figures that contain all of the rigorous statistical analysis for each data set presented in the composite figures, as shown in the following revised figure captions for Figures 1E and 3F:

“Figure 1E) A combined plot enabling comparison of viability, proliferation and wound healing assay for all engineered viral infections of ZR-75, MiaPaCa-2, BxPC-3 and PANC-1 cells. Detailed statistical calculations of each of these experiments were shown in supplementary figures (all viability assays in S Fig. 7-14, wound healing assay in S Fig. 15 and proliferation assay in S Fig. 16 A-C).”

“Figure 3F) A combined plot enabling comparison of the HMGA1 protein levels (right y-axis) and HMGA1 mRNA transcript and NAT transcript levels (left y axis) for all virus infected cell lines. Since the NAT copy numbers were much lower than mRNA copy numbers, the NAT values were multiplied by 10 (for 5.1 NAT position) or 100 (for 3.1 and 4.1 NAT positions) before plotting to facilitate comparison. Detailed statistical calculation of each experiment of this combined plot is shown in S Fig. 22, 25-28.”

6. Introduction:

• To further strengthen the context of this work within current advances in gene therapy and viral vector engineering, the authors may wish to cite the following recent study in the Introduction or Discussion sections:

Nature Biotechnology (2024). 10.1038/s41587-024-02453-3

Including this reference would help readers appreciate the broader landscape of viral vector applications in oncology."

Response – To address this point, we have added the following sentence at the end of the 2nd paragraph in the Introduction section and added the suggested citation:

“It is worth noting that recent advances in viral vector engineering are providing powerful new tools for studying gene therapy approaches and its application in oncology research [9].”

• Briefly define "conditional replication" of adenoviruses for non-specialist readers.

Response – To address this point, we have added the following text in the first paragraph of the Conclusions section

“It should be noted that the background for the engineered viruses examined in this manuscript are adenovirus strains that were previously engineered to be “conditionally replicative” so that they can only replicate in cancer cells. This conditional replication activity originates from deletion of the 19-kDa and 55-kDa E1B genes [23-25]. The 55-kDa E1B protein inactivates the tumor suppressor protein p53, which is often mutated or disabled in cancer cells, but functional in normal healthy cells. Therefore, adenovirus strains lacking the 55-kDa E1B gene are incapable of replicating in normal healthy cells but capable of replicating and lysing cancer cells that lack functional p53. Deletion of the 19-kDa E1B protein, which blocks the downstream effects of p53 to further prevent apoptosis, provides a secondary stringency for conditional replication in cancer cells only.”

Minor Comments:

1. Figure Clarity:

• Figure 1B: Label wound healing images with time points (0h, 24h, 48h) for clarity.

Response – These labels have been added as suggested.

• Figure 3C-D: Include a schematic of the NAT detection strategy in the main text to aid interpretation.

Response – It wasn’t clear how to add this schematic in the absence of using a figure, so we kept the schematic in the figure as Fig 3C, but we added the following text to clarify the explanation of the schematic and to clarify how the PCR-based NAT detection scheme was designed and used.

“Unlike coding strand mRNA that has its introns spliced out, NATs may also contain the intron regions. The HMGA1 gene contains six introns and exons (Fig 3C). To probe for a potential cis-acting HMGA1 NAT, potentially including its introns, we designed a strand-specific RT-primer walk strategy with PCR primers spanning each intron-exon junction (Fig 3C). As the HMGA1 NAT mRNA would be oriented in the opposite direction as the HMGA1 coding strand, the strand-specific RT primers were designed downstream (on the putative NAT mRNA strand) of each intron-exon junction. The PCR primers used to detect the intron-exon junctions are listed in S Table 1 and indicated in Fig 3C. In the absence of genomic DNA (gDNA), an experiment conducted with the PCR primers but in the absence of the RT enzyme would be expected to produce no PCR products. A primer walk experiment using PANC-1 cells is shown in Fig 3D. The negative control (NC) experiment without reverse transcriptase but including the intron-exon junction PCR primer pairs produced no bands indicating that the isolated total RNA was free from gDNA contamination. After this observation, random hexamer RT primers were used to maximize PCR products. A second negative control experiment was performed in which the RT primers were included in the reaction, but the PCR primers were omitted. In this case some PCR primer-independent DNA synthesis was observed for some fragments, apparently as a result of self-priming, which has been reported previously in the literature [15]. Finally, the RT-PCR experiment was performed with product-specific RT primers and with random hexamer RT primers, and strong RT-PCR products were detected at all intron-exon junctions indicating the presence a putative cis-acting HMGA1 NAT.”

2. Discussion Limitations:

• Address potential off-target effects of the adenoviral vectors (e.g., immune responses, unintended gene disruption) and how these might impact clinical translation.

Response – To address this point, we added the following sentence and new citation at the end of the last paragraph in the Conclusions section:

“Several limitations for translation of use of engineered adenovirus vectors to treat human cancers to clinical application have been discussed, including pre-existing immunity in humans, off-target effects such as inflammatory responses and off-target gene expression effects, sequestering to the liver and spleen and immunodominance of the vector genes over transgenes [27].”

3. Writing Edits:

• Abstract: Replace "~25% & 50" with "approximately 25% &50" for consistency.

Response – Replaced as suggested.

Additional Editor Comments:

1. The study is entirely in vitro (cell lines only). While the in vitro results are promising, no in vivo experiments (e.g., mouse tumor models) were performed to validate the efficacy or safety of the adenoviral treatments in an organismal context. This limits the translational relevance of the findings.

Response – Acknowledged. To address this point, the following text has been added to the final paragraph of the Conclusions section:

“While the in vitro results presented here are promising, no in vivo experiments have yet been performed to validate the efficacy or safety of the treatments in an organismal context, which limits the translational relevance of the findings at this point. Next steps for potential translation to clinical use would include in vivo validation, for example, the use of xenograft mouse model studies to assess the efficacy, biodistribution and toxicity in the context of a physiological study.”

2. While decreased HMGA1 protein levels correlate with impaired cancer cell viability and migration, the precise molecular pathways downstream of HMGA1 sequestration or suppression are not investigated. The following related reference could be added to strengthen the manuscript: doi: 10.3389/fmed.2021.756988.

Response – Good suggestion. We have added the following sentence to the first paragraph of the Conclusions section to address this point:

“An example of how downstream pathway activation can be correlated to cancer cell behavior was reported in a recent of a related HMGB1 protein, where it was shown stimulation of exogenous HMGB1, which is ubiquitous in the tumor microenvironment and plays a vital role in tumor recurrence, was directly related to up-regulation of pathways promoting tumorigenesis and down-regulation of pathways suppressing tumorigenesis [23].”

3. In the introduction part, the authors should emphasize the dire need of diagnosis pancreatic cancer at an early stage and following references could be added: doi: 10.4251/wjgo.v16.i4.1256; doi: 10.2147/IJN.S335588.

Response –Agreed. To address this point, we have added the following text at the beginning of the 2nd paragraph of the introduction and added the two suggested references:

“Elevated HMGA1 levels play a critical role in pancreatic cancer tumorigenesis [6]. Pancreatic cancer remains a devastating disease with poor survival statistics and there remains a dire need for improved diagnostic techniques for early detection [7, 8].”

4. Although the viruses were said to have minimal cytotoxicity in healthy epithelial cells (H6c7, HMEC), more rigorous tests (such as assessing normal cell proliferation and gene expression post-treatment) would be needed to confirm safety.

Response –Acknowledged. To address this point, the following text has been added to 1st paragraph of the Conclusions section:

“The assays indicated that all three engineered viruses reduced cancer cell properties, including cell viability, cell proliferation and cell migration capability, all while causing minimal cell toxicity, which is an important factor when considering the safety for potential downstream clinical application. However, more rigorous testing of cell proliferation and gene expression effects post-treatment would be necessary to confirm safety.”

5. The manuscript suggests that the HBS virus is "most effective" without fully acknowledging that all findings are in vitro and that translation to clinical relevance is still speculative. Strong claims about therapeutic application are premature without animal model data.

Response – Acknowledged. To emphasize this point, we modified the last sentence of the abstract as follows:

“The HBS virus designed to sequester HMGA1 proved most effective in suppressing HMGA1 oncogenic activity in these in vitro cell-based studies compared to the AAT and shRNA viruses designed to suppress new synthesis of HMGA1 translation.”

Further, we had already added the following text to the final paragraph of the Conclusions section, which also addresses this point:

“While the in vitro results presented here are promising, no in vivo experiments have yet been performed to validate the efficacy or safety of the treatments in an organismal context, which limits the translational relevance of the findings at this point. Next steps for potential translation to clinical use would include in vivo validation, for example, the use of xenograft mouse model studies to assess the efficacy, biodistribution and toxicity in the context of a physiological study.”

6. Treatment with the AAT virus increased HMGA1 mRNA levels but still decreased HMGA1 protein levels.

Response – Acknowledged. This point had been raised by the reviewer, and we had addressed the point by including by adding the last sentence to this paragraph in the Overview of trends between HMGA1 protein levels, HMGA1 mRNA transcript levels and HMGA1 NAT levels section:

“Finally, a rather su

---

## [Decision Letter · Decision Letter 1]

22 Jul 2025

Dear Dr. Kennedy,

Thank you for submitting your manuscript to PLOS ONE. After careful consideration, we feel that it has merit but does not fully meet PLOS ONE’s publication criteria as it currently stands. Therefore, we invite you to submit a revised version of the manuscript that addresses the points raised during the review process.

We look forward to receiving your revised manuscript.

Kind regards,

Mohammed S. Razzaque, MBBS, PhD

Academic Editor

PLOS ONE

Journal Requirements:

Additional Editor Comments :

The study presents a mechanistically rich body of work, supported by several types of experiments. It is a novel approach, using engineered adenoviruses to modulate HMGA1 expression is of translational relevance. While the experimental depth is admirable, the presentation of the data and the narrative structure of the manuscript could be substantially improved. The readers may find it difficult to follow the central scientific outcome due to redundancy and repetition across multiple sections (e.g., introduction, methods, results, and conclusion). A clearer and more cohesive story would enhance the impact of the findings. It is also unclear about the rationale for including only one breast cancer cell line compared to three pancreatic cancer lines, as this has not been highlighted in the manuscript.

Additionally, there are labeling inconsistency. For instance, “MIA PaCa-2” in the text vs. “Mia-no virus” in Figure 5, requires clarification. Similarly, all the figures (if doable at this stage) would require higher resolution and better annotation. Also, Table 1 should separate the three pancreatic cell lines to maintain consistency with other cell lines presented, as these cell lines were combined in one column, therefore, clarification is needed regarding the reported standard deviations (SD), whether they represent technical replicates or biological replicates across different cell lines.

Lastly, the conclusion section is too lengthy and reiterates much of the discussion. A more concise, one-paragraph conclusion that clearly articulates the key findings and implications would improve readability. The authors might consider integrating portions of the current conclusion into the results and discussion section, which might be helpful to achieve this balance.

Again, the article is too long, and often it is difficult to follow the flow. Authors might consider making it concise as recommended above.

Reviewers' comments:

Reviewer's Responses to Questions

**Comments to the Author**

Reviewer #1: All comments have been addressed

2. Is the manuscript technically sound, and do the data support the conclusions?

Reviewer #1: Yes

3. Has the statistical analysis been performed appropriately and rigorously?

Reviewer #1: Yes

4. Have the authors made all data underlying the findings in their manuscript fully available?

Reviewer #1: Yes

5. Is the manuscript presented in an intelligible fashion and written in standard English?

Reviewer #1: Yes

Reviewer #1: Thank you for your thorough revisions and thoughtful responses to my previous comments. You have comprehensively addressed all concerns, and the manuscript has been significantly improved in terms of clarity, rigor, and overall quality. I appreciate the additional data, expanded explanations, and careful attention to detail in refining the text.

At this stage, I have no further comments or requests for modifications, the manuscript now meets the journal’s publication standards in its current form.

**Do you want your identity to be public for this peer review?** For information about this choice, including consent withdrawal, please see our Privacy Policy

Reviewer #1: **Yes: ** Narsimha Mamidi

---

## [Author Response · Author response to Decision Letter 2]

12 Sep 2025

Below, we address the remaining Reviewer’s and Editor’s comments.

Reviewer’s comments:

The reviewers had no further comments or suggestions regarding the revised manuscript.

Editor’s comments:

1. The study presents a mechanistically rich body of work, supported by several types of experiments. It is a novel approach, using engineered adenoviruses to modulate HMGA1 expression is of translational relevance. While the experimental depth is admirable, the presentation of the data and the narrative structure of the manuscript could be substantially improved. The readers may find it difficult to follow the central scientific outcome due to redundancy and repetition across multiple sections (e.g., introduction, methods, results, and conclusion). A clearer and more cohesive story would enhance the impact of the findings. It is also unclear about the rationale for including only one breast cancer cell line compared to three pancreatic cancer lines, as this has not been highlighted in the manuscript.

Response: The entire manuscript has been carefully edited in an attempt to reduce redundancy throughout the text, to make the language tighter and clearer, and to remove a repetitive summary of the results in the conclusion section. This has resulted in a shortening of the manuscript by ~1.5 pages. Hopefully the result is a revised manuscript that is much easier to read and understand. While our main focus of research has been studying the effect on pancreatic cancer cells, we had one additional breast cancer cell line available in our lab, and we thought that testing and reporting the data for at least one other non-pancreatic cancer cell line would be supportive of the general effectiveness of our approach being tested. We added the following text to clarify this rationale:

“Here, the efficacies for reducing cancer cell characteristics of the two new engineered viruses (AAT and shRNA) were evaluated and compared with the HMGA1 sequestration virus (HBS) in three different human pancreatic cancer cell lines (primary focus) and one human breast cancer cell line (to test generality with one non pancreatic cancer cell line) using in vitro viability, toxicity and necrosis assays.”

2. Additionally, there are labeling inconsistency. For instance, “MIA PaCa-2” in the text vs. “Mia-no virus” in Figure 5, requires clarification. Similarly, all the figures (if doable at this stage) would require higher resolution and better annotation. Also, Table 1 should separate the three pancreatic cell lines to maintain consistency with other cell lines presented, as these cell lines were combined in one column, therefore, clarification is needed regarding the reported standard deviations (SD), whether they represent technical replicates or biological replicates across different cell lines.

Response: The shorthand notation for the cell lines and treatments has been explained in the revised Figure 5 caption. Figures 4 and 5 have been replaced with new figures at high-resolution. Table 1 has been edited to separate the three pancreatic cancer cell lines as recommended. The values in parentheses have been explained in the Table footnote to be the average of three technical replicates performed on each of three biological replicates.

3. Lastly, the conclusion section is too lengthy and reiterates much of the discussion. A more concise, one-paragraph conclusion that clearly articulates the key findings and implications would improve readability. The authors might consider integrating portions of the current conclusion into the results and discussion section, which might be helpful to achieve this balance.

Response: All of the repetitive summary of the data results has been removed from the conclusions section. We have kept four short paragraphs that summarize the results, hypothesize about potential mechanisms, and address important issues for translation to clinical application not addressed in the other sections of the paper. This resulted in a 1-page reduction in the length of the conclusions section.

---

## [Decision Letter · Decision Letter 2]

9 Oct 2025

Dear Dr. Kennedy,

Thank you for submitting your manuscript to PLOS ONE. After careful consideration, we feel that it has merit but does not fully meet PLOS ONE’s publication criteria as it currently stands. Therefore, we invite you to submit a revised version of the manuscript that addresses the points raised during the review process.

Please submit your revised manuscript up to Nov 23 2025 11:59PM. If you will need more time than this to complete your revisions, please reply to this message or contact the journal office at plosone@plos.org . A rebuttal letter that responds to each point raised by the academic editor and reviewer(s). You should upload this letter as a separate file labeled 'Response to Reviewers'.A marked-up copy of your manuscript that highlights changes made to the original version. You should upload this as a separate file labeled 'Revised Manuscript with Track Changes'.An unmarked version of your revised paper without tracked changes. You should upload this as a separate file labeled 'Manuscript'.

We look forward to receiving your revised manuscript.

Kind regards,

Antonio Palumbo jr

Academic Editor

PLOS ONE

Journal Requirements:

Reviewers' comments:

Reviewer's Responses to Questions

**Comments to the Author**

Reviewer #1: All comments have been addressed

Reviewer #3: (No Response)

2. Is the manuscript technically sound, and do the data support the conclusions?

Reviewer #1: Yes

Reviewer #3: Yes

3. Has the statistical analysis been performed appropriately and rigorously?

Reviewer #1: Yes

Reviewer #3: Yes

4. Have the authors made all data underlying the findings in their manuscript fully available?

Reviewer #1: Yes

Reviewer #3: Yes

5. Is the manuscript presented in an intelligible fashion and written in standard English?

Reviewer #1: Yes

Reviewer #3: Yes

Reviewer #1: The authors have fully addressed all of my comments, and the manuscript is now much clearer, more rigorous, and of higher overall quality. I appreciate the inclusion of additional data, expanded explanations, and thorough attention to detail during the revision process. I have no further comments or requests for changes; the manuscript, in its current form, meets publication standards.

Reviewer #3: The authors describe a novel adenoviral approach to target the oncogenic transcription factor HMGA1, which contributes to growth, migration, and chemoresistance in a variety of cancers. Three recombinant adenoviruses (HBS, AAT, and shRNA) targeting HMGA1 by sequestration, artificial antisense transcript, and shRNA expression, respectively, were evaluated in Pancreatic (MiaPaCa-2, PANC-1, BXPC3) and Breast (ZR-75) cancer derived cell lines. The data demonstrate that the HBS virus most potently downregulates HMGA1 and reduces cancer cell viability and migration with insignificant cytotoxic effects in normal cells. The existence of a putative HMGA1 NAT is a new and exciting finding in the HMGA1 regulation field. The manuscript is well-written, and the experiments are well performed and informative. However, some conceptual and contextual refinement would make it more convincing.

Major points

The evidence indicates that apoptosis is triggered by HMGA1 sequestration, but the molecular details of this induction are not known. Downstream targets (p53, BAX, BCL2, SMAD-dependent genes) may be responsible for this effect and may further strengthen the discussion.

The discovery of a cis-acting HMGA1 NAT is a significant result, but any functional consequence is speculation. The authors might discuss whether this NAT functions to stabilize HMGA1 mRNA, prevent miRNA binding or influence translation efficiency, like in other NAT–mRNA systems. A deeper mechanistic insight would be nice they just said in their paper that it will be done in the next paper.

The HBS virus, which was more effective than the AAT and shRNA viruses, sequestered rather than transcriptionally suppressed, yet the discussion does not adequately explain why sequestration is apparently more effective.

Although the study is well done in vitro, it would be nicer if there were a more ambitious statement about how these might be taken forward, perhaps for in vivo or translational work. Specifically, the inclusion of results from xenograft testing, immune response evaluation, and biodistribution studies would have signaled the attention to clinical development problems.

Minor points

Ensure that all figures include clear axis labels, time points, and statistical indications.

The manuscript is well written, but could be slightly condensed in the Results section by minimizing methodological repetition already described in Methods.

**Do you want your identity to be public for this peer review?** For information about this choice, including consent withdrawal, please see our Privacy Policy

Reviewer #1: **Yes: ** Narsimha Mamidi

Reviewer #3: No

---

## [Author Response · Author response to Decision Letter 3]

16 Oct 2025

Below, we address the remaining Reviewer’s comments.

Reviewer #1: This reviewer had no further comments or requests for changes.

Reviewer #3: This reviewer made the following comments:

Major points

1. The evidence indicates that apoptosis is triggered by HMGA1 sequestration, but the molecular details of this induction are not known. Downstream targets (p53, BAX, BCL2, SMAD-dependent genes) may be responsible for this effect and may further strengthen the discussion.

Response: This was an excellent suggestion. We believe that the following revision to the manuscript significantly improves this section. We have added the following text and appropriate references at the end of the “Virus-mediated cell death…” section to discuss how virus-mediated decreases in HMGA1 levels could stimulate increased apoptosis:

“While there is evidence for triggering of apoptosis by all three engineered viruses, the molecular details for this induction are not known. All three viruses cause reduction in the cellular levels of HMGA1. Therefore, one can consider the potential downstream effects that lead to increased apoptosis upon reduction of HMGA1 levels. For example, HMGA1 is known to inhibit the p53 tumor suppressor protein [16, 17], thus blocking its ability to trigger apoptosis, therefore reductions in HMGA1 levels could stimulate increased apoptosis. Furthermore, HMGA1 has been shown to interact directly with p53 in a manner that blocks binding to the Bax promoter, which is a p53 and p21waf1 effectors, further blocking the tumor suppressor function of p53 [18]. It has also been reported that elevated levels of HMGA1 cause upregulation of Il-15 and Il-15Ra, which causes anti-apoptotic effects by inducing expression of apoptosis inhibitors Bcl2/Bcl1/Bcl-x(L) [19-21].”

2. The discovery of a cis-acting HMGA1 NAT is a significant result, but any functional consequence is speculation. The authors might discuss whether this NAT functions to stabilize HMGA1 mRNA, prevent miRNA binding or influence translation efficiency, like in other NAT–mRNA systems. A deeper mechanistic insight would be nice they just said in their paper that it will be done in the next paper.

Response: This point was partly addressed in the section entitled “The role of NATs in regulating HMGA1 transcript stability”. We have elaborated on our previous discussion by adding the following text and appropriate references:

“Therefore, cis-NATs may stabilize HMGA1 mRNA transcripts through cytoplasmic interaction of the NAT with the complementary single-stranded region of the mRNA sequence through interactions of the network of mRNAs, AS transcripts, micro-RNAs and RNA-binding proteins [27]. Another study has shown that mRNA stabilization by NATs can be mediated by cytoplasmic interaction of NATs with single-stranded regions of mRNAs that blocks microRNA destabilization of the mRNA [27, 28]. HMGA1 knock-down or overexpression experiments could be conducted to clarify the mechanism by which HMGA1 NATs stabilize the HMGA1 mRNA or modulate its translation.”

3. The HBS virus, which was more effective than the AAT and shRNA viruses, sequestered rather than transcriptionally suppressed, yet the discussion does not adequately explain why sequestration is apparently more effective.

Response: This result was surprising to us, but the observation was consistent across all experiments and across multiple cell lines. Determination of the underlying mechanism to understand this observation will require additional follow-up studies designed to probe the relative effectiveness of HMGA1-sequestration and reduced HMGA1 transcription. To address this point, we have added more text discussing potential mechanisms by which HMGA1 sequestration may so effective in the following paragraph in the Conclusions section, including addition of appropriate references:

“ As infection with HBS virus exhibited the largest reduction of HMGA1 protein levels and caused significantly more cell death than the other viruses, we concluded that sequestration of HMGA1 protein had a higher impact on suppressing HMGA1 oncogenesis than blocking new synthesis of HMGA1. The precise mechanism by which HMGA1 sequestration reduced HMGA1 protein levels and caused increased cancer cell death remains unclear. The interpretation of the data presented here is complicated by many factors. For example, cells infected with a MOI of 50 ppc deliver 300 HMGA1 binding sites, a number that significantly outnumbers the copy numbers of the HMGA1 mRNA transcripts per cell for every cancer cell type studied. However, the number of protein molecules per mRNA transcript varies significantly by gene [37] and depends on many factors in addition to the number of mRNA transcripts per cell including mRNA stability, the speed of translation and the rate of protein degradation [38]. Another factor that must be considered is that most of the measurements were made at 72-96 hours after virus infection, which would permit at most a single doubling time (MiaPaCa-2 – 40 hrs, BxPC-3 48-60 hrs, Panc-1: 26-53 hrs and Zr-75: 80 hrs) under standard culture conditions. Therefore, any reduced expression due to shRNA or AAT virus infection would not block that action of the already expressed HMGA1, which would, however, be impacted by the sequestration strategy. Another factor that may have influenced the results is the relative accessibility of HMGA1 to the engineered virus HMGA1 binding sites relative to the accessibility of the genomic target HMGA1 binding sites that may be restricted in the context of the chromatin structure [39]. Further experiments may clarify these points, such as side-by-side comparison of chromatin accessibility or HMGA1-DNA binding using a ChIP-seq experiment could provide mechanistic insight into whether the HBS disrupted HMGA1’s interaction with chromatin more effectively than the ATT/shRNA viruses blocked HMGA1 mRNA translation. Based on the observation that HMGA1 sequestration resulted in reduced HMGA1 protein levels, we consider the possibility that an autoregulatory negative feedback mechanism may be involved in regulating HMGA1 levels in cancer cells. Increased cell death in cells with reduced HMGA1 levels could involve a chromatin remodeling mechanism, which HMGA1 is known to use in regulating gene expression, potentially activating apoptotic pathways that are ineffective in the presence of high HMGA1 levels. Further research is needed to determine the complex process by which HMGA1 sequestration reduces HMGA1 protein levels and causes cancer cell death.”

4. Although the study is well done in vitro, it would be nicer if there were a more ambitious statement about how these might be taken forward, perhaps for in vivo or translational work. Specifically, the inclusion of results from xenograft testing, immune response evaluation, and biodistribution studies would have signaled the attention to clinical development problems.

Response: We believe that these points have already been addressed in the final paragraph of the Conclusions section, with the text as follows:

“While the in vitro results presented here are promising, in vivo experiments have not yet been performed to validate the efficacy or safety of the approach in an organismal context, which limits the translational relevance of the findings at this point. Next steps for potential translation to clinical use include in vivo validation, for example, the use of xenograft mouse model studies to assess the efficacy, biodistribution and toxicity in the context of a physiological study. Further obstacles that must be addressed prior to clinical use include pre-existing immunity in humans, characterization of off-target effects such as inflammatory responses and off-target gene expression effects, and sequestration to the liver and spleen and immunodominance of the vector genes over transgenes [40].”

Minor points

5. Ensure that all figures include clear axis labels, time points, and statistical indications.

Response: We believe that this has been addressed.

6. The manuscript is well written, but could be slightly condensed in the Results section by minimizing methodological repetition already described in Methods.

Response: Repetition was eliminated as requested, either by eliminating text in the Methods section or in the Results section.

---

## [Editor Report · Decision Letter 3]

20 Oct 2025

Sequestration and suppressed synthesis of oncogenic HMGA1 using engineered adenoviruses decreases human pancreatic and breast cancer cell characteristics

PONE-D-25-05256R3

Dear Dr. Kennedy,

We’re pleased to inform you that your manuscript has been judged scientifically suitable for publication and will be formally accepted for publication once it meets all outstanding technical requirements.

Kind regards,

Antonio Palumbo jr

Academic Editor

PLOS ONE
---

## [Editor Report · Acceptance letter]

PONE-D-25-05256R3

PLOS ONE

Dear Dr. Kennedy,

I'm pleased to inform you that your manuscript has been deemed suitable for publication in PLOS ONE. Congratulations! Your manuscript is now being handed over to our production team.

Kind regards,

on behalf of

Dr. Antonio Palumbo jr

Academic Editor

PLOS ONE